# LOGOS: LLM-driven End-to-End Grounded Theory Development and Schema Induction for Qualitative Research

## Abstract

Grounded theory offers deep insights from qualitative data, but its reliance on expert-intensive manual coding presents a major scalability bottleneck. Existing computational tools either fail on full automation or lack flexible schema construction. We introduce LOGOS, a novel, end-to-end framework that fully automates the grounded theory workflow, transforming raw text into a structured, hierarchical theory. LOGOS integrates LLM-driven coding, semantic clustering, graph reasoning, and a novel iterative refinement process to build highly reusable codebooks. To ensure fair comparison, we also introduce a principled 5-dimensional metric and a train-test split protocol for standardized, unbiased evaluation. Across five diverse corpora, LOGOS consistently outperforms strong baselines and achieves a remarkable average $80.4\%$ alignment with an expert-developed schema on complex datasets. LOGOS demonstrates a potential to democratize and scale qualitative research without sacrificing theoretical nuance.

## 1 Introduction

Grounded theory (GT) is a methodology for developing theories that emerge directly from empirical data rather than being imposed by prior frameworks (Glaser & Strauss, 2017a). Unlike deductive approaches that begin with hypotheses, GT follows an inductive process—constructing conceptual categories and theoretical explanations from the bottom up. This makes it particularly effective for uncovering latent mechanisms, hidden structures, and recurring patterns in complex phenomena that resist formalization. Historically, GT has produced influential theories across education, management, psychology, communication, politics, sociology, and many other fields (Glaser & Strauss, 2017b; Corbin & Strauss, 1988; Corbin, 2003; Gioia & Chittipeddi, 1991).

GT comprises three core processes: (1) Open coding, which involves freely tagging properties of each datapoint.[1] (2) Axial coding, where codes are merged into semantic clusters and refined for reusability. (3) Selective coding, where relationships among codes are organized into a cohesive theoretical framework. Schema induction (Gick & Holyoak, 1983), originating in cognitive science, describes a similar process of identifying recurring structural patterns across problem instances. In this paper, we use "grounded theory development" and "schema induction" interchangeably.

From a methodological perspective, computer scientists—especially in natural language processing and computer vision—often engage in *informal* grounded theory practices. This typically shows up in the "qualitative analysis" or "error categorization" subsections that appear near the end of experiment sections, even if researchers do not explicitly acknowledge it as such. To produce this section, researchers typically need to manually examine a remarkable number of pairs of input-output to discover the hidden common patterns. From these discoveries, they propose a formalized category or standard where individual observations can fit comfortably into (e.g., error categories, improvement patterns). One of the most formal grounded theory studies is Cemri et al. (2025), which produces great insights into reasons why **multi-agent system fails**. In other tasks, such as robustness-related pitfall identifications (Ribeiro et al., 2020), question decomposition (Wolfson et al., 2020), and ontologies of NLP reasoning tasks (Pi et al., 2022b), grounded theory alike approaches also generate

---

[1]"Code" here refers to the descriptive tag assigned to a datapoint.

fruitful outcome. Typically, computer scientists first derive the conceptual categories and theoretical structure through extensive manual iteration, and only later convert this understanding in to a coded automated annotation workflow.

Despite the desirability of the grounded theory approach, practical costs can be quite expensive. It usually requires a group of experts to read through the corpus and manually perform open, axial, and selective coding. For example, Cemri et al. (2025) deploys 6 experts, each spent more than 20 hours only to mannually examine and code a small 200 execution traces dataset from 7 multi-agent frameworks. Significantly more time is required for code merging, codebook cleanup, conflict resolution, and theorization. Existing attempts at computer-aided grounded theory remain partial and limited. Commercial tools accelerate retrieval and organization but stop short of interpretive synthesis (e.g.,NVivo, MAXQDA). Academic systems, while pushing automation further, still presuppose expert supervision at each critical juncture - validating emergent codes, distilling implicit meanings, and establishing codebook topologies (Gao et al., 2023; Lam et al., 2024; Gao et al., 2025). Even the most recent frameworks provide valuable scaffolds yet still fall short of delivering a fully autonomous, domain-agnostic pipeline. This leaves a persistent gap between the promise of scalable grounded theory and the reality of high expert labor costs.

To empower the qualitative research processes in computer science and broader humanistic and social science researchers, we strive to automate the grounded theory development processes to make it accessible for everyone. In this paper, we introduce LOGOS, a general-purpose, domain-agnostic, and end-to-end solution for grounded theory development. LOGOS combines a relation-aware conceptual graph over codes (with four semantic relations and deductive closure) with an iterative refinement loop that optimizes code abstraction and reusability, yielding the a faithful end-to-end automation of iterative open $\rightarrow$ axial $\rightarrow$ selective coding. In line with grounded-theory practice, LOGOS first induces a codebook from raw text and then reuses this learned schema deductively to code new datapoints and held-out data.

We devise a standardizable statistical metric without any reliance on human evaluation for fair comparison of qualities of different end-to-end generated codebooks when expert codebook is unavailable. Specifically, we propose a novel $5-$dimensional measurement scale following the most principled desiderata of codebooks, which highlights codebook **reusability, semantic fittness, semantic coverage, parsimony, and consistency**. The other side of our novelty is the way we apply our statistical metric: unlike previous approaches (e.g. (Lam et al., 2024; Gao et al., 2025)) that essentially overfit the codebook on the entire corpus before evaluation, we perform train-test split, fitting the codebook on the train set, and evaluate only on the test set. Overall, from a statistical learning perspective, we contribute a evaluation methodology with less statistical bias in codebook quality estimation. Under this standarized metric, LOGOS surpasses $4$ other competent end-to-end coding-based and RAG-based schema induction workflows with a remarkable margin on five datasets from diverse domains. LOGOS also demonstrates excellent human-alignment, with a **surprisingly high expert-schema matching rate of average $80.4\%$** on the Multi-agent System Failure corpus from Cemri et al. (2025) **across various configurations**, validated by human mannual verification.

## 2 RATIONALE AND CONCEPTUALIZATION

**Problem Formulation.** Given a research question $Q$ and qualitative corpus $D = \{d_1, \ldots, d_N\}$ (e.g., interview transcripts, feedbacks, operation traces, et), grounded theory development is the construction process of a *codebook* $\Sigma$ to (i) name recurring structures and phenomena, (ii) organize them into a coherent relational structure, and (iii) support a question-focused report.

Codebook $\Sigma = (C, R)$, where $C = \{c_1, \ldots, c_K\}$ denotes the set of codes. Each code is a short descriptive label (e.g., "frustration with UI", "team communication breakdown") that tags a segment of data with its salient property for later analysis. Much like the knowledge graph formalism, we model code–code relations as a *typed* ternary relation

$$R \subseteq C \times \{\text{sub}, \text{sup}, \text{eq}, \text{orth}, \text{none}\} \times C,$$

so that each element of $R$ is a relation triple $(c_i, r, c_j)$ indicating that code $c_i$ stands in relation type $r$ to code $c_j$. For brevity, we write $(c_i, \text{sub}, c_j) \in R$ as $c_i \rightarrow c_j$ to denote semantic subsumption ("is-a", a partial order), $(c_i, \text{eq}, c_j) \in R$ as $c_i \leftrightarrow c_j$ to denote semantic equivalence, and $(c_i, \text{orth}, c_j) \in R$ as $c_i \perp c_j$ to denote orthogonality (qualitatively disjoint phenomena). The resulting codebook

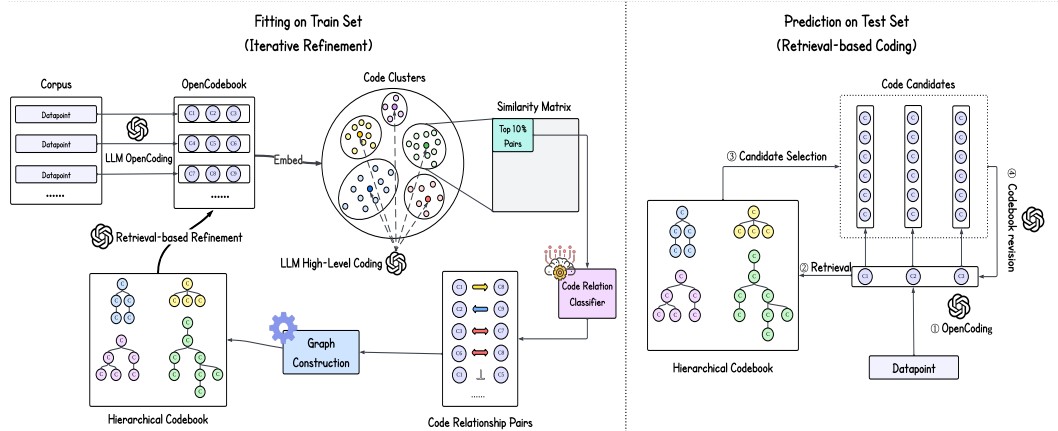

Figure 1: The codebook fitting and prediction mechanism in the `LOGOS` workflow overview. Retrieval based refinement during training shares the same workflow as retrieval based coding.

organizes $C$ into a structured semantic hierarchy enabling downstream querying and theory building.

**5 Highest Desiderata of Codebooks.** When evaluating the qualitative research results, researchers focus on how well the codebook represents the corpus based on the research question (Tracy, 2010; Charmaz & Thornberg, 2021). This makes traditional evaluation methods for LLMs, such as perplexity, unsuitable. We therefore introduce five metrics inspired by established evaluation criteria in qualitative research (Nowell et al., 2017; Kyngäs et al., 2019) and recent computational research (Qiao et al., 2025): (1) *reusable*, allowing codes to generalize across many datapoints and support flexible combinations (e.g., A AND B but not C); (2) *accurate*, faithfully capturing recurring patterns in the data; (3) *comprehensive*, covering all important aspects of the datapoints and the corpus; (4) *parsimonious*, minimizing the semantic overlap between codes in the codebook; and (5) *consistent*, applying reliably across present and future datapoints with similar distributions.

**The Necessity of Datapoint-level Interpretation.** Code generation is essentially an interpretation process of the datapoint. Consider the case where our goal is to identify failure patterns and causations of a multi-agent system. Each datapoint in our corpus is a task execution trace from diverse domains: software engineering, data analyis, Kaggle compeition, interaction with everyday apps and services, scientific reasoning, and more. Naive clustering based on the semantic embeddings make datapoints from similar task domains, rather than the traces with similar failure reasons, to be pushed together. Failure patterns that should have been put into the same category, such as inter-agent miscoordination, communication protocol errors, and task stopping criterion misidentificaiton issues, are not appearing in the same cluster. Hence, attempting to induce a common failure pattern from the *semantic clusters* will likely result in the "garbage in, garbage out" situation. The interpretation process, which can be viewed as a non-linear transformation of a datapoint's syntactic, semantic, and pragmatic attributes (in this case, identifying a datapoint's failure reason and abstracting it out from the datapoint-specific task domains), is usually inevitable for complex grounded theory development. Heavy reasoning, advanced natural language understanding, and systematic expertise external to the datapoint might be required for performing interpretation, necessitating an LLM to perform such non-trivial transformations.

## 3 METHOD

### 3.1 THE LOGOS FRAMEWORK

Upon the problem conceptualization and formulation, we describe `LOGOS`, our implementational-level realization of automated grounded theory development, combining an inductive codebook construction phase with a subsequent deductive coding phase.

**Step 1 - Chunking & Open Coding.** Given a corpus and a research question Q, we divide the text into chunks of 2048 words, with an overlap of 200 words between consecutive chunks. Then we apply `Qwen3-32B` to generate 20 codes conditioned on the research question Q. We carefully design our prompt so that the generated codes reflect the key insights of the document while remain-

ing non-overlapping and unambiguous. This maps to the open coding process in common grounded theory development.

**Step 2 - Embedding & Clustering.** We then embed the codes with `Qwen3-embed-0.6B` and use K-means with a Silhouette and variance score-based selection of K, to gather the codes into semantic clusters. We apply mini-batch K-means (batch size = 1000) for larger codebooks.

**Step 3 - High-level Code Generation** After generating the open codes and the clustering step, and we prompt a LLM to generate $k$ high-level code that describes the gist of the cluster. We add all such high-level codes into the codebook, which later shall be discriminated by the code classifier. Step 2 and 3 together corresponds the axiel coding procedure

**Step 4 - Code Pair Relation Classification.** Inside each cluster, we perform a code merging process (corresponding to axial coding). Conceptually, there are 4 possible relationships for an arbitrary pair of codes $A$ and $B$: (1) $A \rightarrow B$: A is *semantically subordinate* to B - e.g. "writing code with cursor", is a semantic subclass of "using AI-assisted programming tool"; (2) $B \rightarrow A$: B is *semantically subordinate* to A, i.e., the reverse direction. (3) $B \leftrightarrow A$: A is *(near) semantically equivalent* to B Pi et al. (2022a). For example, "building mutual trust with patients" v.s. "cultivating rapport in the patient–provider relationship" are highly equivalent and can be merged into one code. Implementationally, we distill a student `Qwen3-4B-base` model from the `Qwen3-32B` teacher model to classify the code pair relationship; (4) $A \perp B$: A is *orthogonal* to B, which means that A is qualitatively different from B, and there is no semantic containment relationship in either direction. We perform LoRA finetuning, with the language modeling head of the `Qwen-4B-base` model replaced to a 4-category classification head. The distilled model is trained on around $500k$ pairs generated from wikipedia articles, where we generate open codes and annotate the code pairs with the teacher model. We report the performance of different settings of the classifier in the Appendix.

For each cluster, we construct an upper triangle pairwise cosine similarity matrix containing all $\frac{n(n-1)}{2}$ unique pairs. This leverages the symmetry of cosine similarity, where the similarity between $(A, B)$ is equal to that between $(B, A)$. From the matrix, we keep only entries scoring between the interval $[0.5, 0.90]$, and then select the top $10\%$ ranking entries from the filtered set. Finally, we apply the distilled model to the filtered pairs. The rest are automatically treated as *orthognal*. With this approach, we reduce the infeasible corpus-level $O(n^2)$ computation cost to cluster-size level $O(m^2)$, where $m << n$.

**Step 5 - Graph Construction.** With the code pair relationships obtained from the previous step, we construct a hierarchical code graph. Starting from an empty matrix[2] with codebook size by codebook size, we iteratively add code pair relationships into the graph. For each relation we add, we also actively *infer* all possible relationships from the existing ones based on 2 inference rules: (1) transitivity rule - from $A \rightarrow B$ and $B \rightarrow C$, we can infer $A \rightarrow C$; (2) equivalence rule - from $X \leftrightarrow Y$ and $Y \leftrightarrow Z$ we can infer $X \leftrightarrow Z$. To implement the active inference, we maintain a queue and carry breadth-first-search (BFS), popping the leading node each time and apply the two deduction rules and add its relatives into the queue. In realistic running, it is possible that the code pair relation classification label conflicts with the infered label. We resolve any conflicts by the deduction-first principle, produced by our inference engine over the classification results. If two deduction results conflicts, then we keep the deduction result from the previous round.

**Step 6 - Codebook Clean-Up.** After we get the classification result, we perform the following cleanup for the previous dirty codebook: (1) First, we *merge* all of the "mutual" pairs by replacing all the pairs in the group with the one with the highest-score

$$\text{Score} = w_n \cdot S_{\text{freq}} + w_i \cdot S_{\text{in-deg}},$$

where $S_{\text{freq}}$ is global frequency (number of datapoints the code appeared in) and $S_{\text{in-deg}}$ is the incoming degree for each code. For example, suppose we have $A \leftrightarrow B \leftrightarrow C \leftrightarrow D$, and $\max_{Score}(A, B, C, D) = D$, then we simply replace all occurrences of $A, B, C$ with $D$ and adjust global frequency and incoming degree correspondingly[3]. (2) Second, we subsume all of the

---

[2]This matrix is distinct from the cosine similarity matrix in **Step 4**; it instead models the four pairwise relationships as classified by the NLI classifier.

[3]We postpone updating the global frequency and incoming degree of existing nodes until the end of each round, to avoid affecting conditions while nodes are still merging.

low-frequency codes (in our case, we filter out all the codes which appear less than 6 times.) into their parent codes if applicable, and drop those orphan codes below the frequency threshold. For example, if we have $A \rightarrow B, freq(A) = 1$, then we will replace code $A$ with $B$.

## 3.2 ITERATIVE CODEBOOK REFINEMENT MECHANISM

After the first iteration, we have a sparse codebook. To make the codebook useful (i.e., datapoints with overlapping features or patterns can be aggregated via shared codes), the goal is to improve the resuability of the codebook. For this goal, we design iterative codebook refinement mechanism that replaces codes from the previous iteration with best-matching alternative in the codebook.

**Workflow.** Each iteration proceeds in four stages: (1) **Candidate Generation**: up to 10 candidates are generated for each of the ($\leq 20$) codes in the previous codebook; (2) **Pool Assembly & Pruning**: all candidates are aggregated, deduplicated, and pruned to $\leq 200$; (3) **LLM-based Revision**: the datapoint and candidate pool are passed to the LLM to propose a refined set of $\leq 20$ codes; (4) **Output Validation**: the result is parsed and validated.

**Candidate Generation.** For a code connected in the relation graph, we retrieve (a) up to 5 semantically similar codes (similarity $\geq 0.6$), ranked by a hybrid score

$$\text{Score} = w_s \cdot S_{\text{semantic}} + w_f \cdot S_{\text{freq-norm}},$$

where $S_{\text{semantic}} = \cos(\mathbf{e}(c), \mathbf{e}(c')) \in [0.6, 1.0]$ is the cosine similarity between the embedding of the current code $c$ and a candidate code $c'$ (using the same `Qwen3-embed-0.6B` encoder), and $S_{\text{freq-norm}}$ is the global frequency of $c'$ (number of datapoints the code appeared in) normalized to $[0, 1]$; and (b) up to 5 graph-based codes (parents, children, siblings), ranked by frequency. For unconnected codes, we retrieve the top 10 scoring codes from the semantic code retrieval process. If fewer candidates exist, we keep the available set.

**Candidate Pool.** Across $\leq 20$ previous codes, this yields at most 200 candidates. After deduplication, we rank by the hybrid score and prune to the top 200. The final pool can be represented either as a flat list or a dictionary keyed by original codes.

**Prompt Design.** The LLM prompt is structured into four sections: (1) role & goal (refine codebook, prioritize reuse); (2) datapoint text; (3) candidate pool ($\leq 200$); (4) instructions requiring a JSON list output of $\leq 20$ codes, marking any novel ones with "NEW:". We impose a $10k$ token constraint.

**Constraints.** The design ensures that: no more than 20 codes per datapoint, no more than 200 candidates, only codes with similarity $\geq 0.6$ are included, and pruning is always controlled by hybrid ranking. Graph retrieval considers only one layer up/down; unconnected codes fall back to semantic retrieval. Edge cases are naturally bounded by these constraints.

**Validation.** To ensure correct iterative refinement process, two conditions must be met: (1) LLM outputs have $\leq 20$ codes and (2) each new code must be correctly mapped to its respective datapoint. Additionally, it is preferred that LLM ouputs contain a mixture of old codes (from previous iteration) and new codes (that LLM identifies as missing to fully make sense of the input data).

## 3.3 STATISTICAL METRICS FOR CODEBOOK QUALITY

Following the 5 principled desiderata of codebook discussed in Section 2, we introduce our computational operalization for each of the dimensions in this subsection. Following the standard train–test-split protocol in machine learning, with codebook fitting done on the train set, we perform *code prediction* on the test set via a retrieval-based selection algorithm similar to the codebook update in the iterative refinement procedure: for each test datapoint we retrieve a candidate pool of existing codes using the same semantic+graph retrieval as in Section 3.2, and an LLM then selects (but is not allowed to invent) up to 20 codes from this pool. In other words, the test set is coded *deductively* using only the train-derived codebook. After that, we calculate the following metrics of the test codebook.

**Reusability.** Defined as the ratio of used codes to all codes:

$$\text{Reusability} = \frac{\#\text{used codes}}{\#\text{all codes}}.$$

Here, *all codes* refers to the size of the final, cleaned codebook learned on the train split (after Steps 4–6 and iterative refinement). A *used* code is any code from this train-derived codebook that is assigned at least once to a test datapoint during the deductive code prediction step described above; codes that remain in the codebook but are never selected on the test set are counted as unused. This measures the practical utility of a codebook on unseen data and ranges in $[0, 1]$.

**Descriptive Fitness.** A score assigned by a large language model (LLM) in the range of 1–10, that *quantifies the extent to which selected codes properly describe a datapoint (even if partially).* For example, describing a resturant review that complains about delivery speed as "parking difficulty" would be considered improper. To keep the scale consistent with other metric dimensions, we rescale the fitness score to $[0, 1]$. Since calculating the semantic fittness for the entire test set might be computationally expensive, we recommend doing sampling based estimation. The same sampling-based scoring applies to calculation of coverage.

**Descriptive Coverage.** Similarly LLM-scored in the range of 1–10, *coverage evaluates to what extend do all the essential aspects of a datapoint relevant to the research question are captured by the assigned codes*. For instance, describing a resturant review that complains both "food serving speed" and "parking difficulty" only as "parking difficulty" misses the other essential side of the complaint, and thus is partial and incomplete. In this sense, a set of code can be proper but partial – two relatively orthognal dimensions. We also rescale coverage score to $[0, 1]$ to keep mangnitude consistency.

**Parsimoniousness.** A structural measure capturing semantic redundancy among codes:

$$\text{Parsimoniousness} = 1 - \frac{2}{n(n-1)} \sum_{i<j} \cos(c_i, c_j),$$

where $n$ is the number of codes and $\cos(c_i, c_j)$ is the cosine similarity between $c_i$ and $c_j$. This penalizes highly similar or redundant code pairs, encouraging compact yet expressive codebooks.

**Consistency / Stability.** To measure generalization across datasets, we compute the Jensen–Shannon Divergence (JSD) between the empirical code distributions on the train and test sets:

$$\text{Stability} = 1 - \text{JSD}(P \,\|\, Q),$$

where $P$ and $Q$ denote the respective distributions. A higher stability score indicates that the codebook maintains consistency across data splits. Notice that this metric is under the assumption that train and test set are independent, identical distribution.

**The Final Composite Metric.** In practice, these individual scores can be combined into a weighted sum to provide a single goodness-of-fit score for the codebook. Metrics reflecting semantic adequacy, such as descriptive fitness and coverage, may be given higher weight relative to structural measures like parsimony and consistency. Since reusability is essential for performing aggregation and multi-view clustering of the dataset, it also makes sense to assign higher weight to reusability as well. For our current work, we use equal weight for simplicity.

## 4 EXPERIMENT AND DISCUSSION

### 4.1 DATASET AND EXPERIMENT SETTING

We evaluate the performance of LOGOS across diverse domains using five datasets: (1) **Ali Abdaal transcripts**: A collection of full-text transcripts collected from YouTube videos by the million-subscriber creator Ali Abdaal. This dataset contains 500 documents with an average length of 4,204 words. (2) **Podcast transcripts**: We adopt the same dataset used by Edge et al. (2024), consisting of transcripts from the podcast *Behind the Tech with Kevin Scott*. The corpus is segmented into text chunks of 2,048 words with a 200-word overlap between consecutive chunks. (3) **Abstracts dataset**: A set of 210 randomly sampled UIST paper abstracts spanning the years 1989–2018 used by Lam et al. (2024). (4) **Multi-Agent System (MAS) failure records**: A dataset of multi-agent

Table 1: Performance of different methods.

| Method | AliAbdaal | Podcast | Abstracts | MAS | Math Failure |
|--------|-----------|---------|-----------|-----|--------------|
| OpenCoding | 2.762 | 2.915 | 2.420 | 2.594 | 2.286 |
| LLOOM | 2.615 | 2.527 | 2.463 | 2.462 | 2.629 |
| GraphRAG | 2.443 | 2.352 | 2.428 | 2.588 | 1.922 |
| LightRAG | 2.352 | 2.408 | 2.091 | 2.568 | 1.956 |
| Thematic-LM | 2.557 | 2.711 | 2.458 | 2.742 | 2.589 |
| HICode | 2.772 | 2.836 | **2.589** | **2.996** | 1.916 |
| Logos (Iter-1) | 2.810 | 2.958 | 2.206 | 2.703 | 2.303 |
| Logos (Best) | **3.002** | **3.169** | 2.495 | 2.831 | **2.647** |

Figure 2: Statistical metrics breakdown.

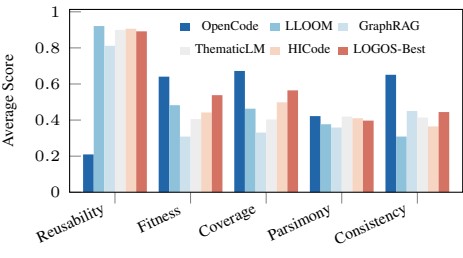

system failure trajectories introduced by Cemri et al. (2025), containing 785 LLM-annotated trajectories with lengths ranging from 1,262 to 71,640 words. We use a subset of 230 programming task datapoints for experiment 1. (5) **Math failure dataset**: Solutions to GSM8K math problems (Cobbe et al., 2021) generated by Qwen-32B. Incorrect solutions were filtered out, yielding 316 records with an average length of 500 words.

## 4.2 EXPERIMENT 1 - STATISTICAL METRICS COMPARISONS

We report the distributional metrics of all approaches (reuseability, fitness, covereage, parsimoniousness, and consistency) introduced in Section 3.3. For all datasets, we report the summed statistical metrics. We choose four competent baselines in our study: (1) **OpenCoding** is the simplest one, where we apply LLM-generated open codeing for each datapoint as the baseline. There is no iteration and no axiel or selective coding. We aggregate the codes to produce the final codebook, and discard codes below a minimum-frequency. (2) **LLOOM** is an reimplementation of the pipeline from Lam et al. (2024) to make it fit onto our datasets. Specifically, we drop the text extraction part, and tune the hyperparameters like number of open code per datapoint and clustering size. We keep the high-level codes generated from the code clusters as the final codebook, and discard the open codes, as the original workflow. (3) **GraphRAG** is the off-the-shelf GraphRAG pipeline. We only change the question prompt to enforce the LLM to produce a 100-item codebook from all communities. (4) **LightRAG** is the off-the-shelf LightRAG pipeline, with similar question adaptation. (5) **Thematic-LM** aggregates codes through sequential LLM-based consolidation, where multiple coder agents produce codes that are merged iteratively by an aggregator and reviewer using an adaptive codebook that continually updates as new data arrive. (6) **HICode** performs batch-level hierarchical clustering, using LLMs to inductively merge and abstract labels in repeated rounds of top-down synthesis, producing progressively higher-level themes without an adaptive, continuously updated codebook. For more details of the baselines, please check Appendix A.7.

From Table 1, LOGOS is consistently the top-performing method. To better understand the performance, we also provide the metric breakdown of each dimension of our proposed statistical metric, shown in Figure 2. From the results, we could observe that OpenCoding and LLOOM are like two extremities – OpenCoding greatly "overfits" to the training set, with a lot of sparse and non-reusable datapoint-specific codes (for example, on Ali Dataset, OpenCode produces 15827.6 unique codes for each research question on average). The global average reusability score for OpenCoding is as low as 0.209, and the global average fittness and coverage for OpenCoding is 0.641 and 0.672, respectively. In contrast, LLOOM leaves only the cluster-specific high-level codes, which delivers very high reusability rate (0.921), at the sacrifice of semantic adequacy (LLOOM has a fittness score of 0.482 and a coverage score of 0.463). In their middle ground, LOGOS nicely reconcile the reusability and semantic adequacy, delivering almost the reusability score (0.892) as LLOOM, without significant sacrifice of semantic fittness (0.538) and coverage (0.565). Besides, we can also clearly observe that GraphRAG and LightRAG are not good at codebook generation without non-trivial modification of the original workflow, especially when the research question requires advanced interpretations of the datapoint. Their reusability is not on par with LLOOM and LOGOS, with significantly worse semantic adequancy. Last, but not least LOGOS maintains high-ranking parsimony and consistency.

## 4.3 EXPERIMENT 2 - EFFECTIVENESS OF LOGOS ITERATIVE REFINEMENT

To understand how much the iterative refinement mechanism helps improve the codebook quality, we run LOGOS for 10 iterations on the **AliAbdaal**, **Podcast**, and **Abstracts** dataset. We show the reusability, descriptive fittness, descriptive coverage, parimoniousness, and consistency scores across iterations as a line plot in Figure 3. As we can see, the reusability rate of the codes gradually rises across iterations, at the cost of gradually descreasing semantic adequacy (i.e., codebook fittness and coverage). This suggests that the sparsity issue of LLM-generated open code gets increasingly

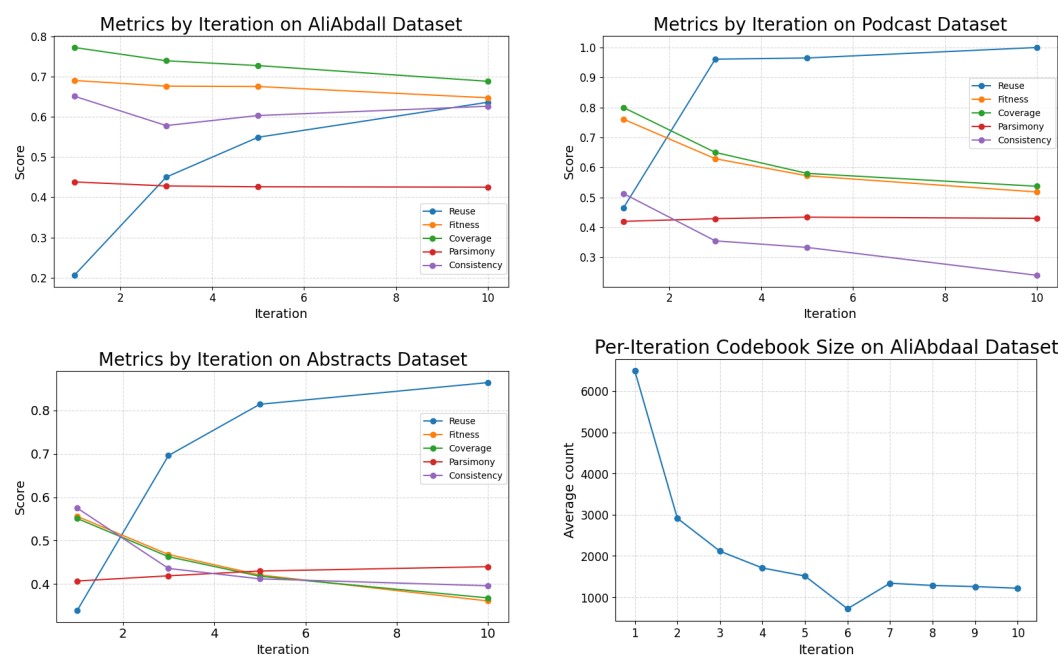

Figure 3: Breakdown of five distributional metrics dimensions and codebook size variations.

betters, with the overly-detailed, datapoint-specific codes gradually replaced by more generalizable and reuseable higher-level codes (the codebook size demonstrates a decreasing trend). Parsimony remains relative stable, whereas consistency demonstrates a decreasing pattern across iterations.

## 4.4 EXPERIMENT 3 - CASE STUDY: GLOBAL SENSE-MAKING QUESTION ANSWERING

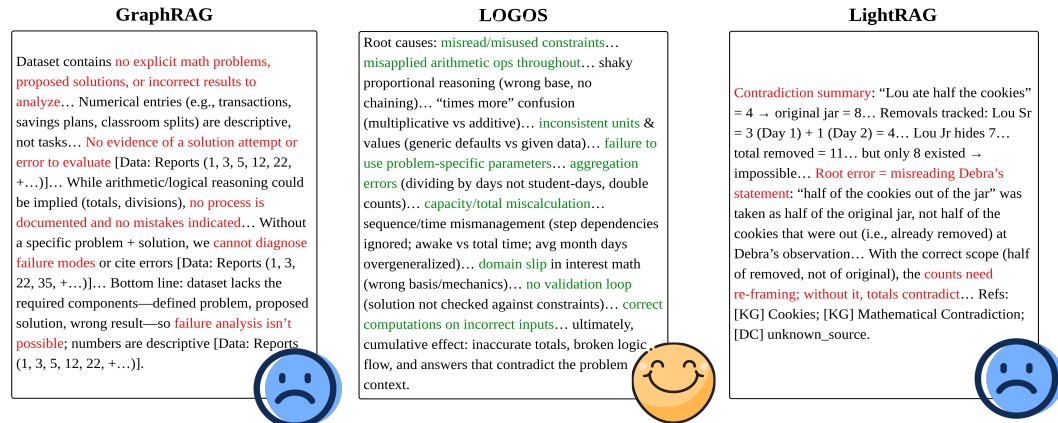

Figure 4: On Math Failure dataset, GraphRAG and LightRAG fails to capture failure errors because no relevant information is extracted. In contrast, LOGOS successfully deliver a sensemaking answer.

GraphRAG and LightRAG studies a new family of questions which they address as "global sense-making questions". Their defining difference is that relevant information for global questions dissipate across the entire corpus and is more than spans and entities. For example, consider the question "What insights do leaders give on allocating budget for tech development?", A single interview only contain one leader's insight for very specific industry sectors and cannot fully answer the question. Dozens of such relevant information need to be combined, integrated, and synthesized into the final answer (open-ended, no ground truth).

Schema induction question can be understood as a subspecies of global-level sensemaking question with special research value. It primarily focuses on *verbally distilling and abstracting the commonalities, structural similarities, and recurring patterns* from the entire corpus.

However, unlike those global information-synthesis questions that can be readily resolved by using entity and relation extraction based knowledge graph , schema induction requires heavy interpretation beyond extraction. To show case the qualitative and paradigmatic limitation of GraphRAG and LightRAG, we experiment on **Math Failure Dataset**, with global question "What are LLMs' failure patterns in mathematical reasoning?") From the results we show in Figure 4, we can easily observe that GraphRAG and LightRAG cannot generate sense-making answer for the research questions at all. After mannually examining the knowledge graph from the two RAG approaches, we find that the reason is that: *failure patterns, which are much more not-readily-extractable logical dependencies, operation procedures, formula choice and value filling, are out of the knowledge graph.* Nodes and edges in the knowledge graphs are primarily about the entities and semantic relationships between them, which could not afford answering the schema-induction question. In constrast, LOGOS successfully identifies the recurring patterns and summarize them into a sense-making final answer.

### 4.5 EXPERIMENT 4 - CASE STUDY: ALIGNMENT WITH HUMAN GROUND TRUTH

Our primary goal of building LOGOS is to liberate mannual labors. To see whether LOGOS can truly reach high agreement with human expert annotators, we compare the LOGOS generated codebook with the expert ground-truth (GT) schema on the **MAS dataset**. After 5 iterations of LOGOS, our codebook produces a fine-grained codebook with size 286 (with a hierarchical structure). Since the GT codebook only has a size of 17, we perform LLM-based aggregation of our codebook to generate 30 clusters. For each clusters, we generate 3 high-level codes that describes the theme of the cluster. After this, we directly dump

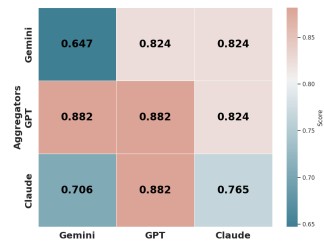

Figure 5: Code match heatmap.

the clusters-codes along with the GT codebook together into a LLM to do matching. For each of the code from the GT codebook, we count as a successful match as long as it matches with at least one of the 30 clusters from LOGOS. The final matching rate could reach **88.2%** under the best configurations (GPT-O3 as aggregator), whereas average performance is still as high as **80.4%**. [4] Since the codebook size is relatively small, we are also able to mannually confirm that the matching results are valid. This suggests that LOGOS has encouraing potentials to emancipate human labors in grounded theory developement and schema induction processes.

## 5 RELATED WORK

**Computer-Aided Grounded Theory Development**  The high cost of qualitative research has motivated a decades-long effort to offload the coding and sense-making pipeline onto software (Chen et al., 2016; Bryda & Costa, 2023). Commercial CAQDAS tools such as NVivo and MAXQDA accelerate retrieval—searching for keywords and bulk-applying a priori codes—but stop short of interpretive work. Academic efforts, especially within the HCI community, move beyond keyword matching toward automation that still foregrounds expert judgment. Cody (Rietz & Maedche, 2021) and PaTAT (Gebreegziabher et al., 2023) serves as a code recommender by learning from patterns in human-generated codes. CoAICoder (Gao et al., 2023) leverages AI to enhance human-to-human collaboration within CQA. These efforts show great potential in AI-assisted qualitative analysis, yet challenges regarding semantic understanding and generative quality remain significant barriers.

Recently, researchers begun leveraging Large Language Models to facilitate qualitative research(Hou et al., 2024; Schroeder et al., 2025; Barros et al., 2025). CollabCoder (Gao et al., 2024), ThemeViz (Kang et al., 2025) and Montes et al. (2025) demonstrate that LLM can scaffold distributed teams, though they primarily treat the LLM as a conversational partner. Building on this, several works (Rao et al., 2025; Sharma & Wallace, 2025; Wiebe et al., 2025; Xu et al., 2025) aim to establish efficient human-AI collaborative frameworks for qualitative analysis. Similarly, Mind-Coder (Gao et al., 2025) introduces a framework designed to produce transparent analytical traces. Despite these advancements, experts generally remain responsible for distilling implicit meanings

---

[4]We provide the complete mapping from cluster names to ground truth codebook in Appendix A.2.

from raw codes. From another branch, researchers explored fully automated qualitative analysis workflow. Paoli (2024) and Khan et al. (2024) perform initial trials on LLM-based automatic thematic analysis(Braun & Clarke, 2006). LLOOM (Lam et al., 2024), Thematic-LM (Qiao et al., 2025), HICode (Zhong et al., 2025) and Auto-TA (Yi et al.) offer end-to-end pipelines that move from an unannotated corpus to a comprehensive codebook. They basically start from a similar pattern: an LLM assigns codes to each datapoint, but then using different aggregation methods to group into a set of high-level themes. Consistent with the thematic-analysis paradigm, their goal is to summarize patterns across the corpus, not to construct an explicit hierarchy of concepts or rigidly model logical relationships among codes (Braun & Clarke, 2021). In contrast, grounded theory is explicitly hierarchical, relational, and iterative. It develops initial multi-level codes through open and axial coding, and integrates them into a rigorous theoretical structure during selective coding(Charmaz, 2006; Glaser & Strauss, 2017a; Charmaz & Thornberg, 2021). The objective is to produce a reusable, persistent, and logically rigid conceptual structure that meaningfully represents the corpus and generalizes to future data. LOGOS advances toward fully automated grounded-theory development by starting from LLM-based open coding and then (1) constructing a hierarchical code graph with explicit relations, (2) iteratively refining the codebook to increase reusability and parsimony, and (3) reusing the learned schema deductively on held-out data. This design mirrors standard grounded-theory practice and enables LOGOS to support much deeper and finer-grained interpretive analyses (e.g. for discursive analysis, interpretative phenomenological analysis, and narrative analysis – see our YC narrative analysis in Appendix A.3.1) in broader qualitative research, beyond just classifying a datapoint into several high-level themes.

**Automated Theoretical Induction for Specialized Domains** Abundant works have been done specifically for fully-automated *event* schema induction in recent years(Li et al., 2020; 2023; 2022; Jin et al., 2022; Du, 2022). The prevailing methodology represents events as traditional knowledge graphs—modeling actions and named entities as nodes with strong inductive biases on edge definitions—and performs reasoning using graph neural networks. Cheng et al. (2024) follow a similar formalism but target the specific domain of the EV battery supply chain. Evaluation metrics in this area typically rely on next-event prediction (e.g., edge type classification). Regarding broader theoretical induction, researchers across various domains have increasingly integrated LLMs into their workflows. For instance, policy researchers have employed LLMs to synthesize themes from vast textual corpora (Fang et al., 2025; Wang et al., 2025). Literature surveys, a fundamental qualitative task, represent another promising avenue for LLM-based analysis, particularly for identifying and organizing themes across extensive collections of published work (Übellacker, 2024; Singh et al., 2025; Padmakumar et al., 2025). Within the computer science community specifically, LLMs have been applied to qualitative analysis tasks in software engineering (Bano et al., 2024; Leça et al., 2025). However, while these approaches attempt to align recurring entities, relations, and actions from individual data points, they rely on strong assumptions regarding domain (e.g., restricted to geopolitical events), ontology (limiting schemas exclusively to events), and topology (adhering to specialized knowledge graph formalisms). `LOGOS` seeks to overcome these constraints, delivering a fully automated, general-purpose schema induction system.

## 6 CONCLUSION

LOGOS closes a long-standing gap between retrieval-heavy tooling and true interpretive synthesis for grounded theory. It delivers an end-to-end, domain-agnostic pipeline which simulates the open → axial → selective coding processes via semantic clustering, relation-aware graphs, and iterative refinement. Together, we deliver a standardizable 5-dimensional codebook quality statistical metric, along with a train/test protocol. Empirically, LOGOS outperforms coding- and RAG-based baselines on 5 corpora and attains $88.2\%$ alignment with expert schemas on MAS, indicating real potential to cut expert labor without discarding nuance. Current limits primarily reside in the limitation of codebook typology, which only consider semantic hierarchy. We plan to consider richer causal/temporal relations in the future, with better computaional cost optimzation (e.g. LLM routing).

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

# A  APPENDIX

## A.1  LIMITATIONS AND ETHICAL CONSIDERATIONS

LOGOS has several limitations. Our current codebook typology only models hierarchical semantic relations (subsumption, equivalence, orthogonality). However, hierarchical semantic relations only model static taxonomic structure (e.g., a dog is an animal). It does not yet capture richer structures such as causal, temporal, or processual relations between codes that can model either a snapshot or full dynamic system that involves temporal state and cyclic update (Simon, 2012). These dynamic relational structures that are the central approach to grounded theory (Braun & Clarke, 2021) which hierarchical semantic relation alone can't achieve. Also, LOGOS is still computationally resource–intensive: it performs multiple LLM passes over the corpus and iterates on the codebook, which may be costly for very large datasets or for researchers without access to substantial compute. Future work will focus on routing, distillation, and other efficiency optimizations.

Ethically, LOGOS inherits the biases and blind spots of the underlying LLMs. The induced codebooks may reflect unfair or stereotypical framings (e.g., gender, race, politics) and misinterpret culturally situated narratives. Also, LOGOS assumes that input corpora have been collected, anonymized, and stored in accordance with relevant consent, privacy, and data-governance norms; these responsibilities remain with the human researchers. Yi et al. provide a novel method to use multi-persona agents for codebook quality and alignement with human analysts. It's worth exploring the multi-agent usage in multi-round codebook iteration. In addition to potential biases, LLM-driven analyses can introduce factual or interpretive errors. LOGOS may occasionally misrepresent the corpus—for instance, by fabricating relations, overgeneralizing themes, or missing context-dependent nuances. These errors arise from known limitations of LLMs, including hallucination and overgeneralization. We therefore view LOGOS as a drafting tool, not an oracle: researchers should critically review, edit, and, where necessary, contest the automatically produced codes and themes.

## A.2  ON EVALUATING SCHEMA ALIGNMENT

The evaluation in Section 4.5 shows high conceptual alignment (88.2%) with the expert-developed ground-truth (GT) schema, which we frame as recall and coverage; although LOGOS produced 30 clusters versus 17 expert codes, a conventional precision metric is both hard to define and methodologically less relevant for grounded theory. The goal is conceptual containment rather than replication: GT is exploratory, and different researchers naturally produce schemas at different abstraction levels. What matters is that expert-identified concepts are captured within LOGOS's more fine-grained, many-to-one mappings—for example, an expert's broad "External Tool Error" may reasonably decompose into clusters such as C20 (Authentication and credential handling errors) and C21 (External API misuse assumptions) within our codebook. "Unmatched" clusters are often a feature, not a bug, surfacing novel insights, finer-grained distinctions, or alternative abstractions that augment human analysis. For this reason, precision is less important: it presumes that unmatched outputs are errors, whereas in exploratory coding surplus concepts add nuance and reveal overlooked patterns. Prioritizing recall/coverage better reflects the purpose of grounded theory—to comprehensively map the conceptual landscape—while penalizing lower precision would mischaracterize the value of detailed discovery.

**Cluster Names**

**C1:** Specification compliance failures

**C2:** Role and responsibility violations

**C3:** Repetition and looping errors

**C4:** Context and memory breakdowns

**C5:** Stop-condition and termination awareness gaps

**C6:** Clarification and information-seeking failures

**C7:** Task focus and objective drift

**C8:** Peer and external input disregarded

**C9:** Reasoning–action inconsistency

**C10:** Premature stopping before validation

**C11:** Missing or skipped verification steps

**C12:** Incorrect or faulty verification methods

**C13:** Quantitative reasoning and calculation errors

**C14:** Constraint and requirement misinterpretations

**C15:** Misunderstanding of fixed facts or premises

**C16:** Handling of incomplete or underspecified data

**C17:** Ethical, legal, and normative reasoning gaps

**C18:** Missing edge cases or special-case handling

**C19:** Debugging, error detection, and self-correction weaknesses

**C20:** Authentication, authorization, and credential handling errors

**C21:** Misuse or incorrect assumptions about external APIs

**C22:** Mishandling of complex data structures and relationships

**C23:** Exception handling and reporting deficiencies

**C24:** Testing and coverage insufficiencies

**C25:** Ambiguity interpretation and resolution weaknesses

**C26:** Overgeneralized or misapplied heuristic reasoning

**C27:** Inadequate rigor in proofs and formal reasoning

**C28:** Weak synthesis or integration of intermediate results

**C29:** Overconfident but incorrect assertions

**C30:** Limited breadth in exploration or search

**FC Categories → Cluster Sets.** *Higher-level FC categories mapped to the broadened cluster set (one-to-many).*

- **FC-1 (Specification Issues)** → C01, C02, C14, C15
- **FC-2 (Inter-Agent Misalignment)** → C06, C07, C08, C09, C25
- **FC-3 (Task Verification)** → C10, C11, C12, C24, C19

**Ground Truth FM Items → Clusters.**

- **FM-1 (Opening Greetings)**
- **FM-2 (Founder Introductions Credentials)** → C09, C01
- **FM-1.3 (Step repetition)** → C03
- **FM-1.4 (Loss of conversation history)** → C04, C16
- **FM-1.5 (Unaware of termination conditions)** → C05, C25
- **FM-2.1 (Conversation reset)** → *missing (no cluster assigned)*
- **FM-2.2 (Fail to ask for clarification)** → C06, C25
- **FM-2.3 (Task derailment)** → C07, C26
- **FM-2.4 (Information withholding)** → *missing (no cluster assigned)*
- **FM-2.5 (Ignored other agent's input)** → C08, C25
- **FM-2.6 (Reasoning–action mismatch)** → C09, C28
- **FM-3.1 (Premature termination)** → C10, C19
- **FM-3.2 (No or incomplete verification)** → C11, C24
- **FM-3.3 (Incorrect verification)** → C12, C13, C27

**Currently Unused Clusters** :

- **C17:** Ethical, legal, and normative reasoning gaps
- **C18:** Missing edge cases or special-case handling
- **C20:** Authentication, authorization, and credential handling errors
- **C21:** Misuse or incorrect assumptions about external APIs
- **C22:** Mishandling of complex data structures and relationships
- **C23:** Exception handling and reporting deficiencies
- **C29:** Overconfident but incorrect assertions
- **C30:** Limited breadth in exploration or search

## A.3 EXTENSION OF HUMAN ALIGNMENT CASE STUDY IN 4.5

Similar to the MAS dataset, we extend the human alignment case study on Vibe Coding dataset Sarkar & Drosos (2025) and a curated set of 10 Y Combinator accepted application YouTube videos, each annotated with ground-truth codes by expert researchers. Our method achieves a $100\%$ on Vibe Coding dataset and $80\%$ matching rate on Y Combinator Dataset.

### A.3.1 Y COMBINATOR ACCEPTED APPLICATION DATASET RESULT:

We curated 10 YouTube videos submitted by startup companies and accepted to Y Combinator. The videos range in length from 45 seconds to 1 minute 37 seconds, with an average transcript length of approximately 237 words.

Included videos are:

1. https://www.youtube.com/watch?v=BBhAJwgTlZ4
2. https://www.youtube.com/watch?v=LlYe-he1knQ
3. https://www.youtube.com/watch?v=q_HIpz3pVRc
4. https://www.youtube.com/watch?v=R6G8GrDSPTU
5. https://www.youtube.com/watch?v=Rzlr2tNSl0U
6. https://www.youtube.com/watch?v=dkOpG3kqmy4
7. https://www.youtube.com/watch?v=BxjmoN6LhqM
8. https://www.youtube.com/watch?v=VGXkghBFtL0
9. https://www.youtube.com/watch?v=VYM-lvMI5QY
10. https://www.youtube.com/watch?v=pw5IObxW8yQ

**Cluster Names**

**C1:** Value Proposition & Solution Messaging

**C2:** Pitch Delivery Style

**C3:** Problem Narrative Construction

**C4:** Problem-Solution Structure

**C5:** Problem Validation through Data

**C6:** Problem Universality & Urgency

**C7:** Competitive Criticism

**C8:** Competitive Differentiation

**C9:** Technical Capability Demonstration

**C10:** Technical Architecture & AI Solution

**C11:** Solution Positioning

       **C12:** Early Traction Signals

       **C13:** Revenue & Monetization Evidence

       **C14:** Metrics & Launch Data

       **C15:** Platform Integration & Partnerships

       **C16:** Market Expansion Vision

       **C17:** YC Direct Engagement

       **C18:** Audience Engagement Techniques

       **C19:** Momentum & Progress Signaling

       **C20:** Founder Credibility

**Ground Truth FM Items → Clusters.**

- **FM-1 (Opening & Greetings)** *missing (no cluster assigned)*
- **FM-2 (Founder Introductions & Credentials)** → C09, C20
- **FM-3 (Problem Definition & Validation)** → C03, C04, C05, C06
- **FM-4 (Competitive Analysis)** → C07, C08
- **FM-5 (Solution & Product)** → C01, C04, C08, C09, C10, C11
- **FM-6 (Traction & Validation)** → C12, C13, C14, C15
- **FM-7 (Target Market & Distribution)** → C11, C15, C16
- **FM-8 (Execution & Progress)** → C02, C14, C18, C19
- **FM-9 (YC-Specific Elements)** → C17
- **FM-10 (Closing & Gratitude)** → *missing (no cluster assigned)*

### A.3.2 VIBE CODING DATASET RESULT:

The original Vibe Coding dataset Sarkar & Drosos (2025) includes 4 Twitch streams and 36 YouTube videos. For our analysis, we focus exclusively on the YouTube portion of the dataset. Of the 36 YouTube videos, 33 were accessible at the time of analysis (the unavailable videos are **Nt2Lkdy3f5Y**, **_QOvocOFLbo**, and **NYVaCr3T1T0**). We use only the transcript up to the duration reported in the original Vibe Coding paper. The average transcript length of these videos is $3,342$ words, with video durations ranging from 2 minutes 35 seconds to 45 minutes 16 seconds.

**Cluster Names**

       **C1:** AI Behavior, Reliability, and Limitations

       **C2:** AI as Co-Creative Partner

       **C3:** AI Output Customization and Control

       **C4:** AI-Driven Content, UI/UX, and Visual Design

       **C5:** AI-Assisted Automation & Development Efficiency

       **C6:** AI Evaluation, Comparative Assessment, and Oversight

       **C7:** Community, Personalization, and Engagement

       **C8:** Integration, Tooling, and Platform Interoperability

**Ground Truth FM Items → Clusters.**

- **FM-1 (Goals)** → C02, C04, C05, C07
- **FM-2 (Intentions)** → C07
- **FM-3 (Workflow)** → C01, C02, C03, C04, C05, C08
- **FM-4 (Prompting)** → C02, C03, C04

- **FM-5 (Debugging)** $\rightarrow$ C01, C06
- **FM-6 (Challenges)** $\rightarrow$ C01, C05, C06, C08
- **FM-7 (Expertise)** $\rightarrow$ C01, C02, C03, C04, C05, C06, C08
- **FM-8 (Trust)** $\rightarrow$ C01, C02, C03, C06, C07
- **FM-9 (Definition of Vibe Coding)** $\rightarrow$ C02, C07

### A.3.3 LLM-AS-A-JUDGE ALIGNMENT WITH HUMAN

To validate the accuracy of our LLM-as-a-judge method regarding description fitness and coverage, we assessed the correlation between human and LLM evaluations. We randomly selected 50 data points from Iteration 3 of LOGOS on the MATH Failure dataset, a subset chosen for its optimal performance in prior experiments. For the human evaluation, two authors manually reviewed the data points and corresponding LOGOS-generated code, rating fitness and coverage on a scale of 0–10. Simultaneously, we employed Qwen3-32B as the evaluative model, prompting it with the math question, correct answer, incorrect answer, and the generated code. We then calculated the Pearson correlation coefficient between the human and LLM scores. The resulting coefficients were 0.7297 for fitness and 0.6952 for coverage, indicating a strong alignment between human and automated judgments.

**Prompt for fitness:** You are evaluating how well the codes accurately describe the reason why the proposed answer is wrong based on the question and correct answer. Score from 1 to 10:

10: All codes perfectly match the content

8-9: Most codes match well, minor issues

6-7: Some codes match, some don't

4-5: Many codes don't match the content

1-3: Most codes don't match the content

**Prompt for coverage:** You are evaluating how well the codes cover the important aspects of the reason why the proposed answer is wrong based on the question and correct answer. . Score from 1 to 10:

10: Codes cover all important aspects

8-9: Codes cover most important aspects

6-7: Codes cover some important aspects

4-5: Codes miss many important aspects

1-3: Codes miss most important aspects

### A.4 THEMATICLM BASELINE REPORT

| Method | AliAbdaal | Podcast | Abstracts | MAS | Math Failure |
|--------|-----------|---------|-----------|-----|--------------|
| Thematic-LM | 2.596 | 2.788 | 2.412 | 2.680 | 2.384 |

### A.5 DISTILLED CLASSIFIER RESULTS

Since the performance of the distilled classifier determinantly impacts the quality and consistency of the constructed graph, we also carry out extensive experiments to improve its performance. We attempted to fine-tune various models with different approaches, and record all results in Table 2. From the results, the general observation is that: (1) doing global fine-tuning results in similar performance as doing LoRA fine-tuning; and (2) Using a larger model significantly improves the performance.

Our Wikipedia training corpus consist of 350k code pairs generated from English Wikipedia, using the same embedding plus taking top30% highest cosine similarities pipeline as in LOGOS. On this

corpus, we observe a serious label imbalance issue. The pairs labeled "mutual" account for only approximately $10\%$ of those with other labels. Therefore, we also attempted the label smoothing technique and the oversampling-based balancing technique. Results show that the balancing technique is effective in our label-imbalance case.

| Model | Acc | F1 Macro | F1 Micro | Balanced Acc |
|---|---|---|---|---|
| MiniLM-L12 (33M) | 0.536 | 0.507 | 0.536 | 0.626 |
| Roberta-focal-loss | 0.722 | 0.654 | 0.722 | 0.734 |
| Modern-Bert-full | 0.723 | 0.665 | 0.723 | 0.694 |
| Roberta-large-full | 0.740 | 0.674 | 0.740 | 0.735 |
| Roberta-MNLI-full | 0.730 | 0.664 | 0.730 | 0.733 |
| Roberta-large-LoRA | 0.719 | 0.649 | 0.728 | 0.728 |
| Qwen3-0.6B | 0.766 | 0.711 | 0.766 | 0.715 |
| Qwen3-4B-it | 0.803 | 0.747 | 0.802 | 0.742 |
| Qwen3-4B-base | 0.804 | 0.745 | 0.804 | 0.759 |
| **Qwen3-4B-balanced** | **0.814** | **0.815** | **0.814** | **0.812** |

Table 2: Model performance comparison across Accuracy, F1 Macro, F1 Micro, and Balanced Accuracy.

All models are trained on a single A100 GPU with using the `HuggingFace` `AutoModelforClassification` framework with the following hyperparameters: Label smoothing was applied with a factor of $0.1$. Whenever applicable, focal loss gamma was set to $2.0$. A weighted sampler was used for class balancing. Training was performed for 10 epochs with a learning rate of $2 \times 10^{-4}$, per-device training batch size of $256$, and per-device evaluation batch size of $512$. The LoRA configuration employed a rank $r = 16$, $\alpha = 32$, dropout rate of $0.05$, bias set to `none`, and targeted modules included `q_proj`, `k_proj`, `v_proj`, and `o_proj`, with `score` modules preserved.

## A.6 CODE FREQUENCIES PER DATASET AND QUESTION

### A.6.1 PAPER ABSTRACTS DATASET QUESTIONS AND TOP 25 FREQUENT CODES:

- **Q1:** What are the problem framings and research gaps identified within the UIST abstracts? This question aims to identify the common ways that authors in the UIST community frame their research. It explores the recurring themes and narratives used to establish the significance of their work, such as addressing technological limitations, filling gaps in existing research, or enabling new forms of interaction. By looking at the collection of abstracts, you can identify the shared understandings of what constitutes a "problem" in this research community.

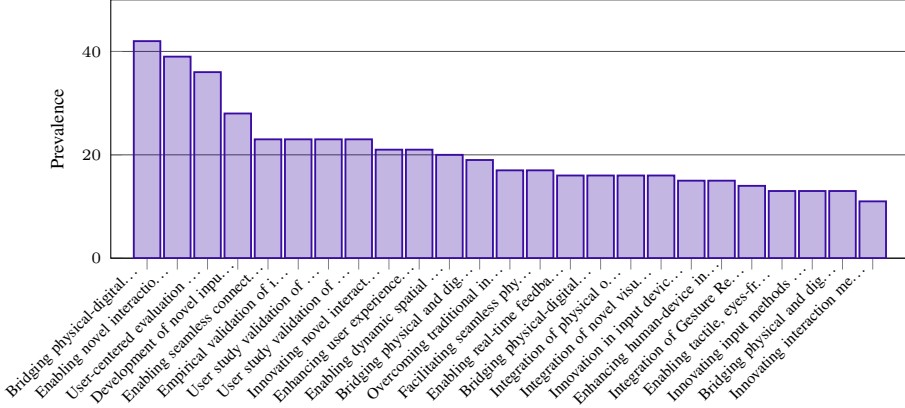

Paper Abstracts - Q1 (Top 25)

- Bridging physical-digital interaction paradigms
- Enabling novel interaction techniques through unconventional inputs
- User-centered evaluation of system effectiveness

- Development of novel input devices for enhanced interaction
- Enabling seamless connection between physical and digital systems
- Empirical validation of interaction techniques
- User study validation of system effectiveness
- User study validation of new interface effectiveness
- Innovating novel interaction techniques
- Enhancing user experience through novel input methods
- Enabling dynamic spatial awareness in interactive systems
- Bridging physical and digital interaction
- Overcoming traditional input device constraints
- Facilitating seamless physical-digital transitions
- Enabling real-time feedback in interactive system design
- Bridging physical-digital interaction
- Integration of physical objects with digital systems
- Integration of novel visualization techniques
- Innovation in input device design for specific tasks
- Enhancing human-device interaction through touch
- Integration of gesture recognition in input devices
- Enabling tactile, eyes-free interactions
- Innovating input methods for diverse applications
- Bridging physical and digital interaction paradigms
- Innovating interaction methods beyond traditional displays

- **Q2:** What are the forms of research contributions claimed across the abstracts? This question seeks to categorize the main contributions presented in the abstracts. It helps in understanding the landscape of research outputs, distinguishing between new systems, interaction techniques, fabrication methods, empirical studies, or theoretical frameworks that are the primary focus of the research. This will map the terrain of innovation at UIST and show where the collective research effort is concentrated.

Paper Abstracts - Q2 (Top 25)

- Real-Time Interaction Techniques
- Design flexibility for user interfaces
- Empirical validation through user studies
- User study-driven performance evaluation
- Integration of physical and virtual interaction
- Design challenges in customizable web interfaces
- Facilitating finger-based, whole-hand, and tangible input methods
- Design implications for two-handed interaction techniques

- Integration of Physical and Virtual Interaction
- Real-Time Feedback Mechanisms in Interactive Systems
- User interface design for parameterizable systems
- Realtime interaction framework development
- Real-time interaction techniques
- Framework for user-defined interaction techniques
- Empirical validation of system performance
- User interface optimization based on actual usage
- Enhanced exploration of complex design spaces via interactive tools
- Interactive physical-digital hybrid systems
- Integration of computer vision in interface design
- Design variations for two-handed input systems
- Development of tangible interaction techniques
- Integration of advanced features in UI toolkits
- System adaptability to different use cases
- Context-aware interaction techniques
- Empirical validation of prototype system performance

- **Q3:** What are the foundational methods that underpin the research in the abstracts? This question focuses on identifying the core method, technological, material, and computational building blocks that enable the presented research. It aims to reveal trends in the use of specific methods, hardware, software approaches, or fabrication techniques that are frequently cited as being central to the innovations described across the dataset.

Paper Abstracts - Q3 (Top 25)

- Empirical validation of research prototypes
- User study validation of system effectiveness
- Development of intuitive interaction techniques
- User-Centered Evaluation of Interaction Techniques
- Human-Computer Interaction (HCI) Research
- System design and implementation
- Innovative interaction technique development
- Development of Intuitive Interaction Techniques
- Reduction of implementation complexity
- User interface prototyping with minimal resources
- Interactive system architecture
- User study for method evaluation
- Gestural input methods for user interaction

- – User experience evaluation
- – Real-time physical-digital interaction synchronization
- – Modular component-based design
- – Real-time interaction between physical and virtual elements
- – Innovative engineering of human-computer interface
- – Empirical validation of interactive design hypotheses
- – Physical-digital integration for interactive design
- – Modular system design for multiple applications
- – Design of input devices for two-handed interaction
- – Design implications for two-handed interaction techniques
- – Real-time interaction processing
- – Modular component-based design for interfaces

- **Q4:** What are the user roles and paradigms of interaction implied in the abstracts? This question investigates how the "user" is conceptualized in the research. It explores the different ways users are expected to interact with the proposed systems and technologies—whether as active creators, performers, collaborators, or passive consumers of information—and what these roles suggest about the underlying assumptions and values of the research. This can reveal the models of human-computer interaction being explored and promoted.

Paper Abstracts - Q4 (Top 25)

- – Real-Time Processing for Interaction
- – Hybrid Physical-Digital Interaction Models
- – Dynamic adaptation to user input
- – Task-oriented interaction design
- – Seamless integration of physical and digital interaction
- – Dynamic Interface Adaptation for User Needs
- – Real-Time Virtual Feedback from Physical Actions
- – Context-aware interaction models
- – Dynamic System Adaptation to User Input
- – User-centered interaction design
- – Innovative input method development
- – User-Centered Spatial Interaction Design
- – User-Centered Interaction Design
- – Hybrid physical-digital interaction models
- – Dynamic interface adaptation for user needs
- – Integration of physical gestures in digital interaction
- – User-centered spatial interaction design

- User study validation of interaction models
- User-Centered Evaluation of System Utility
- Physical-digital hybrid interaction models
- Collaborative Multi-User Interaction
- Seamless integration of physical and digital spaces
- Support for exploratory design experimentation
- User-Centric Spatial Interaction Design
- Seamless Integration of Physical and Digital Spaces

- **Q5:** What are the evaluation strategies and validation methods employed to demonstrate the contributions? This question aims to understand how the UIST community validates its research claims. It focuses on identifying the methodologies used to evaluate novel systems and techniques, such as controlled user studies, technical benchmarks, expert reviews, or illustrative case studies. Analyzing these methods will shed light on the community's standards of evidence and what is considered a robust demonstration of a research contribution.

Paper Abstracts - Q5 (Top 25)

- Technical validation through practical implementation
- Technical demonstration of system concepts
- Technical feasibility demonstration through prototyping
- Technical validation through system functionality
- Prototype-based system demonstration
- Empirical validation through user studies
- Implementation-based validation through usage scenarios
- Integration of multiple functionalities into a unified interface
- Design of multi-touch interaction techniques
- Dynamic user interface as a medium for interaction
- Design space exploration for novel interaction paradigms
- Hybrid physical-digital interaction frameworks
- Use of controlled user studies for evaluation
- Technical validation through implementation
- Technical innovation through interaction design
- Illustrative case study demonstration of system capabilities
- User-Centered Interface Design
- Technical implementation of visualization system

- Evaluation through implementation
- User interface as a dynamic medium for interaction
- Demonstration of practical applicability
- Empirical validation of system performance
- User interface innovation assessment
- Technical benchmarking against existing techniques
- Research contribution through tool creation

### A.6.2 PODCAST DATASET QUESTIONS AND TOP 25 FREQUENT CODES:

- **Q1:** How are different technological and scientific domains characterized and interconnected within the conversations? This question seeks to understand the mental models of the tech landscape presented by the guests. Rather than just listing topics, it focuses on the language used to describe various fields, the relationships drawn between them (e.g., AI and biology), and which areas are framed as foundational, emerging, or peripheral. This reveals the conceptual structure of the tech world as seen by its leaders.

Podcast - Q1 (Top 25)

- – Interdisciplinary convergence in technological innovation
- – Synthesis of personal passion and professional purpose in tech careers
- – Cultural impact of democratizing technical knowledge
- – Interdisciplinary Synergy in Innovation
- – Democratization of technical capabilities
- – Empowerment through hands-on technical skills
- – Interdisciplinary problem-solving as a catalyst for innovation
- – Legacy of foundational tech pioneers shaping modern innovation
- – Democratization of technical knowledge and skills
- – Interdisciplinary collaboration as catalyst for innovation
- – Interdisciplinary problem-solving through cross-domain thinking
- – Interdisciplinary convergence in AI development
- – Resilience through iterative learning and failure
- – Need-Driven Adoption of Emerging Technologies
- – Interdisciplinary integration in technological innovation
- – Mentorship shaping career trajectories
- – Democratization of technological creation
- – Interdisciplinary convergence of technology and creative expression
- – Learning through failure and iteration

– Interdisciplinary synergy in innovation
– Interplay between technical and creative skills
– Mentorship and peer influence in tech career paths
– Interplay between technical mastery and creative problem-solving
– Democratization of programming and tech access
– Historical context of tech innovation

• **Q2:** What are the narratives and visions for the future of technology and science as articulated by the guests? This question aims to identify the recurring ways in which thought leaders speculate about and envision the future. It explores the common themes, hopes, and anxieties they express regarding long-term technological trajectories and their potential societal impact. The goal is to synthesize the collective imagination and forecasting present across the interviews, looking for shared optimistic or cautionary tales.

Podcast - Q2 (Top 25)

– Interdisciplinary approaches to problem-solving
– Interdisciplinary collaboration shaping technological progress
– Interdisciplinary collaboration in technological advancement
– Balancing innovation with societal responsibility
– Long-term societal impact of emerging technologies
– Interdisciplinary approaches to tech challenges
– Resilience Through Iterative Learning and Adaptation
– Long-term societal implications of emerging technologies
– Ethical considerations in AI development and application
– Interdisciplinary approach to tech and creativity
– Curiosity as a driver of technological exploration
– Ethical considerations in AI development and deployment
– Interplay between technological progress and evolving societal values
– Interdisciplinary Synergy in Tech Innovation
– Interplay between personal passion and professional trajectory
– Long-term societal impact of technological innovation
– Long-term societal impact of AI advancements
– Long-term societal implications of AI advancements
– Long-term societal implications of AI integration
– Societal implications of advancing AI capabilities
– Ethical frameworks for responsible AI development
– Interdisciplinary approaches to AI development
– Human-AI collaboration as a cornerstone of future progress

– Democratizing access to advanced technology
– Interplay between personal passion and professional innovation

- **Q3:** What are the ways technological and societal challenges are framed? This question investigates the shared understanding of 'what's broken' and 'how to fix it.' It focuses on identifying patterns in how problems—whether technical, ethical, or social—are defined. The analysis should capture who or what is held responsible for these challenges (e.g., market forces, lack of regulation, historical inertia).

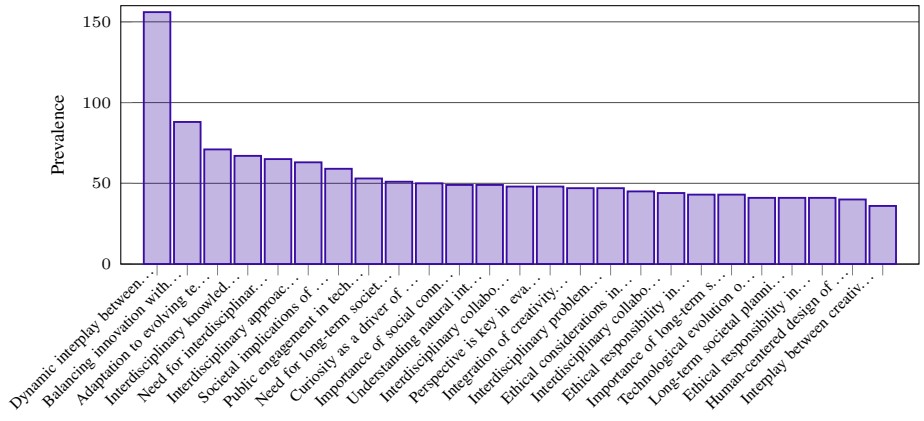

Podcast - Q3 (Top 25)

– Dynamic interplay between creativity and technical execution
– Balancing innovation with human-centric values
– Adaptation to evolving technology platforms
– Interdisciplinary knowledge integration for problem-solving
– Need for interdisciplinary collaboration in problem-solving
– Interdisciplinary approaches to problem-solving
– Societal implications of AI adoption
– Public engagement in tech ethics
– Need for long-term societal planning
– Curiosity as a driver of innovation
– Importance of social connection in technology
– Understanding natural intelligence to inform machine design
– Interdisciplinary collaboration as problem-solving strategy
– Perspective is key in evaluating tech challenges
– Integration of creativity in technical fields
– Interdisciplinary problem-solving approaches
– Ethical considerations in technological development
– Interdisciplinary collaboration in AI research
– Ethical responsibility in technological development
– Importance of long-term societal planning
– Technological evolution outpaces societal adaptation
– Long-term societal planning for technological impact
– Ethical responsibility in AI development
– Human-centered design of AI systems
– Interplay between creativity and technical execution

- **Q4:** How do the guests conceptualize the ecosystem of technological innovation, particularly the interplay between academic research, industry development, and commercial application? This question explores the different models of progress and innovation discussed in the podcast. It aims to uncover the various perspectives on how new ideas are

born, developed, and scaled. The analysis should focus on the perceived roles, tensions, and synergies between fundamental research in academia and applied work in industry, revealing the underlying philosophies about how technological advancement happens.

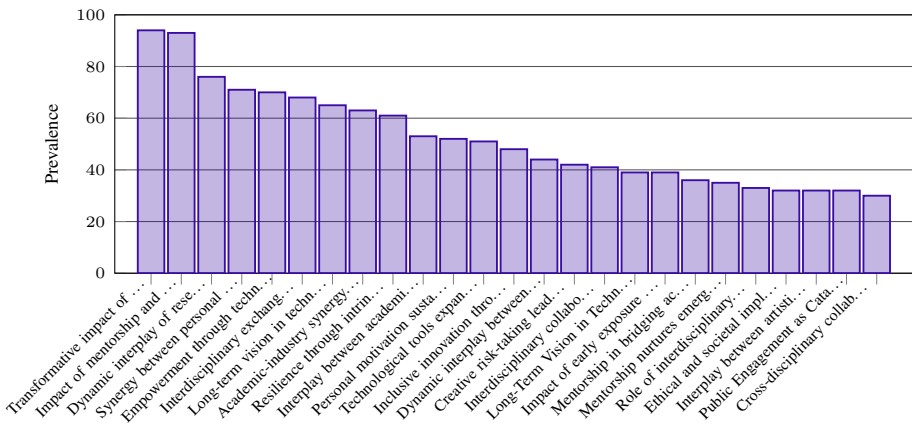

Podcast - Q4 (Top 25)

- Transformative impact of mentorship and visibility
- Impact of mentorship and peer interaction
- Dynamic interplay of research and application
- Synergy between personal passion and professional purpose
- Empowerment through technical literacy and creativity
- Interdisciplinary exchange fuels breakthroughs
- Long-term vision in technological development
- Academic-industry synergy in innovation
- Resilience through intrinsic motivation
- Interplay between academic research and industry application
- Personal motivation sustains long-term innovation efforts
- Technological tools expand creative possibilities
- Inclusive innovation through grassroots initiatives
- Dynamic interplay between art and engineering
- Creative risk-taking leads to innovation
- Interdisciplinary collaboration in innovation
- Long-Term Vision in Technological Development
- Impact of early exposure to technology
- Mentorship in bridging academic and professional gaps
- Mentorship nurtures emerging talent
- Role of interdisciplinary perspectives in innovation
- Ethical and societal implications of technological advancement
- Interplay between artistic vision and technical feasibility
- Public Engagement as Catalyst for Scientific Relevance
- Cross-disciplinary collaboration for systemic solutions

- **Q5:** What are the personal and professional motivations that drive these thought leaders? This question seeks to build a composite picture of the values and driving forces behind the work of leaders in science and technology. It involves analyzing the stories guests tell about their careers, their passions, and what they find meaningful. This can reveal a "taxonomy of purpose," identifying threads like intellectual curiosity, entrepreneurial ambition, humanitarian goals, or a desire to build elegant systems.

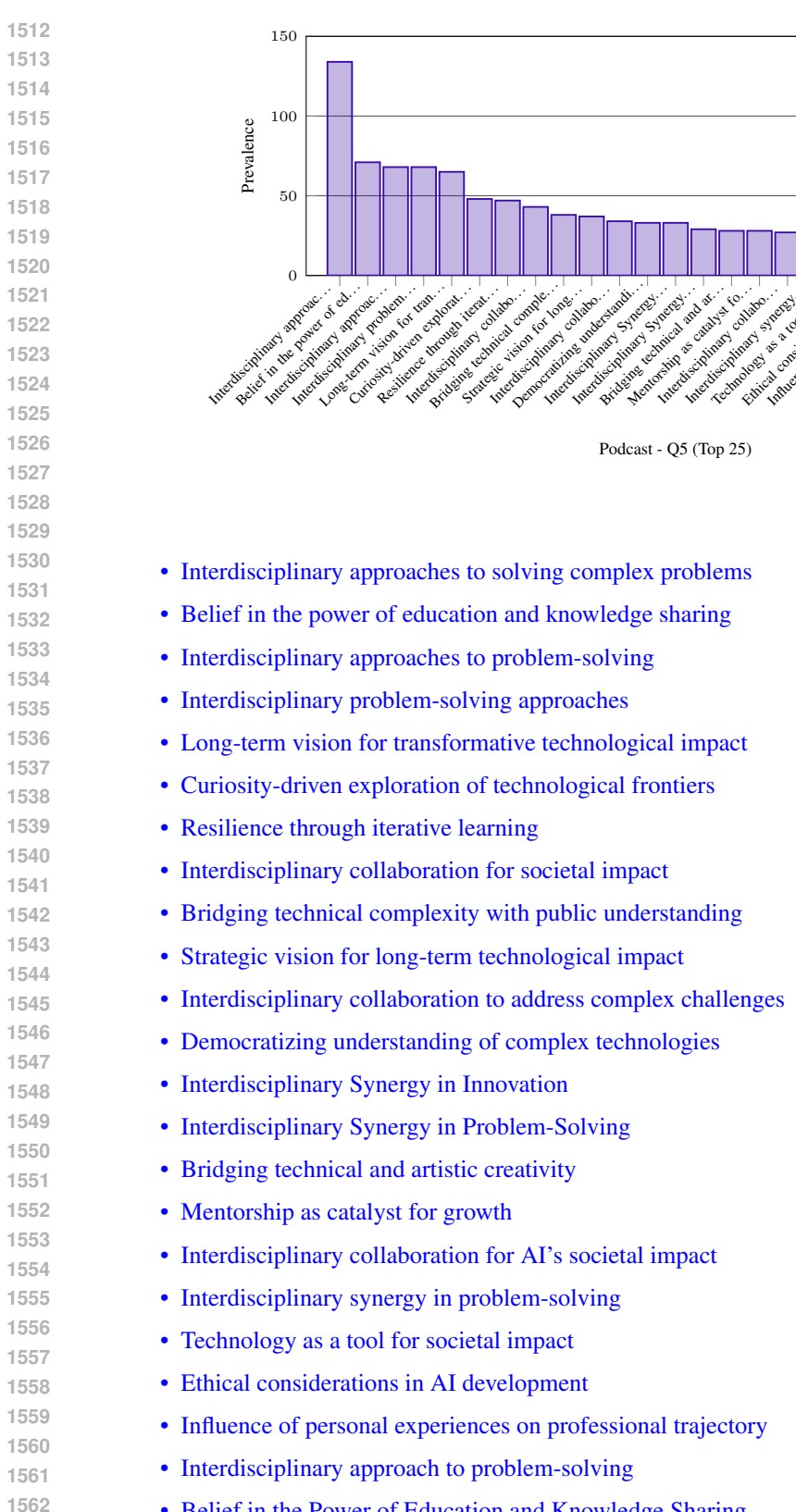

Podcast - Q5 (Top 25)

- Interdisciplinary approaches to solving complex problems
- Belief in the power of education and knowledge sharing
- Interdisciplinary approaches to problem-solving
- Interdisciplinary problem-solving approaches
- Long-term vision for transformative technological impact
- Curiosity-driven exploration of technological frontiers
- Resilience through iterative learning
- Interdisciplinary collaboration for societal impact
- Bridging technical complexity with public understanding
- Strategic vision for long-term technological impact
- Interdisciplinary collaboration to address complex challenges
- Democratizing understanding of complex technologies
- Interdisciplinary Synergy in Innovation
- Interdisciplinary Synergy in Problem-Solving
- Bridging technical and artistic creativity
- Mentorship as catalyst for growth
- Interdisciplinary collaboration for AI's societal impact
- Interdisciplinary synergy in problem-solving
- Technology as a tool for societal impact
- Ethical considerations in AI development
- Influence of personal experiences on professional trajectory
- Interdisciplinary approach to problem-solving
- Belief in the Power of Education and Knowledge Sharing
- Bridging creative passions with technical skills
- Democratizing access to knowledge and skills

### A.6.3 ALIABDAAL DATASET QUESTIONS AND TOP 25 FREQUENT CODES:

- **Q1:** What core motivations or needs seem to drive the speaker's messages across multiple videos? This question asks deeper needs or motivations the speaker articulates or implies repeatedly throughout the dataset. It is not about cataloging explicit reasons (e.g., "productivity tools"), but uncovering fundamental drivers like autonomy, personal growth, or emotional reassurance. It involves inductively coding references to why ideas are presented, and comparing across episodes to surface an emergent overarching category.

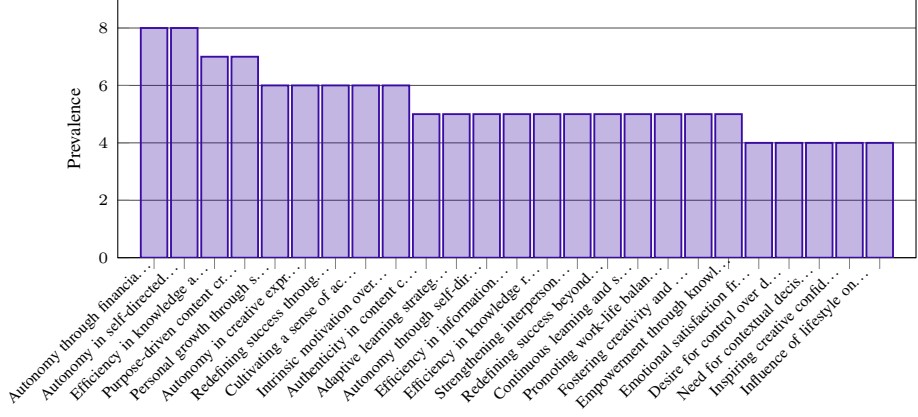

AliAbdaal - Q1 (Top 25)

  - Autonomy through financial independence
  - Autonomy in self-directed learning
  - Efficiency in knowledge acquisition
  - Purpose-driven content creation
  - Personal growth through skill acquisition
  - Autonomy in creative expression
  - Redefining success through personal fulfillment
  - Cultivating a sense of accomplishment
  - Intrinsic motivation over external validation
  - Authenticity in content creation
  - Adaptive learning strategies
  - Autonomy through self-directed learning
  - Efficiency in information processing
  - Efficiency in knowledge retention
  - Strengthening interpersonal relationships
  - Redefining success beyond traditional metrics
  - Continuous learning and skill development
  - Promoting work-life balance
  - Fostering creativity and artistic expression
  - Empowerment through knowledge sharing
  - Emotional satisfaction from device use
  - Desire for control over digital environment
  - Need for contextual decision-making in tech
  - Inspiring creative confidence and exploration
  - Influence of lifestyle on device preference

- **Q2:** How does the speaker construct authority and credibility in their narrative across the videos? This question explores how the speaker positions themselves as trustworthy or knowledgeable—not by counting statements, but by examining patterns such as personal anecdotes, credentials, data citations, or collaborative language. It asks: "What strategies

recur that help the speaker convey credibility?" The goal is to develop a conceptual category of how credibility is built conversationally across the dataset.

AliAbdaal - Q2 (Top 25)

- Leveraging personal experience as credibility anchor
- Validating audience's struggles through shared experiences
- Emphasizing practical application of theoretical concepts
- Leveraging personal journey as credibility anchor
- Utilizing personal experience as credibility anchor
- Offering actionable, practical advice
- Creating a supportive community through shared goals
- Using relatable personal anecdotes to build trust
- Promoting actionable advice for practical application
- Actionable advice with real-world examples
- Highlighting practical application of theoretical concepts
- Positioning self as continuous learner
- Constructs credibility through problem-solution frameworks
- Leveraging personal experience as a credibility anchor
- Positioning self as a continuous learner
- Utilizing metaphors to simplify complex concepts
- Highlighting real-world application of theoretical concepts
- Validating audience struggles through shared experiences
- Utilizing relatable personal anecdotes to build trust
- Validating audience experiences through shared struggles
- Positioning Oneself as a Continuous Learner
- Validating audience concerns through shared experiences
- Encouraging incremental progress over perfection
- Storytelling to illustrate real-life impact
- Using metaphors to simplify complex concepts

- **Q3:** What conceptual framing does the speaker consistently apply to challenges or obstacles discussed? This question invites identification of how the speaker discusses struggles—not as isolated events, but through recurring conceptual lenses like "growth," "process," "inevitable learning curves," or "systemic support." It asks: "What unifying conceptual frame does the speaker apply to adversity?" Analysis across transcripts should surface a category that explains the general interpretive approach to challenges.

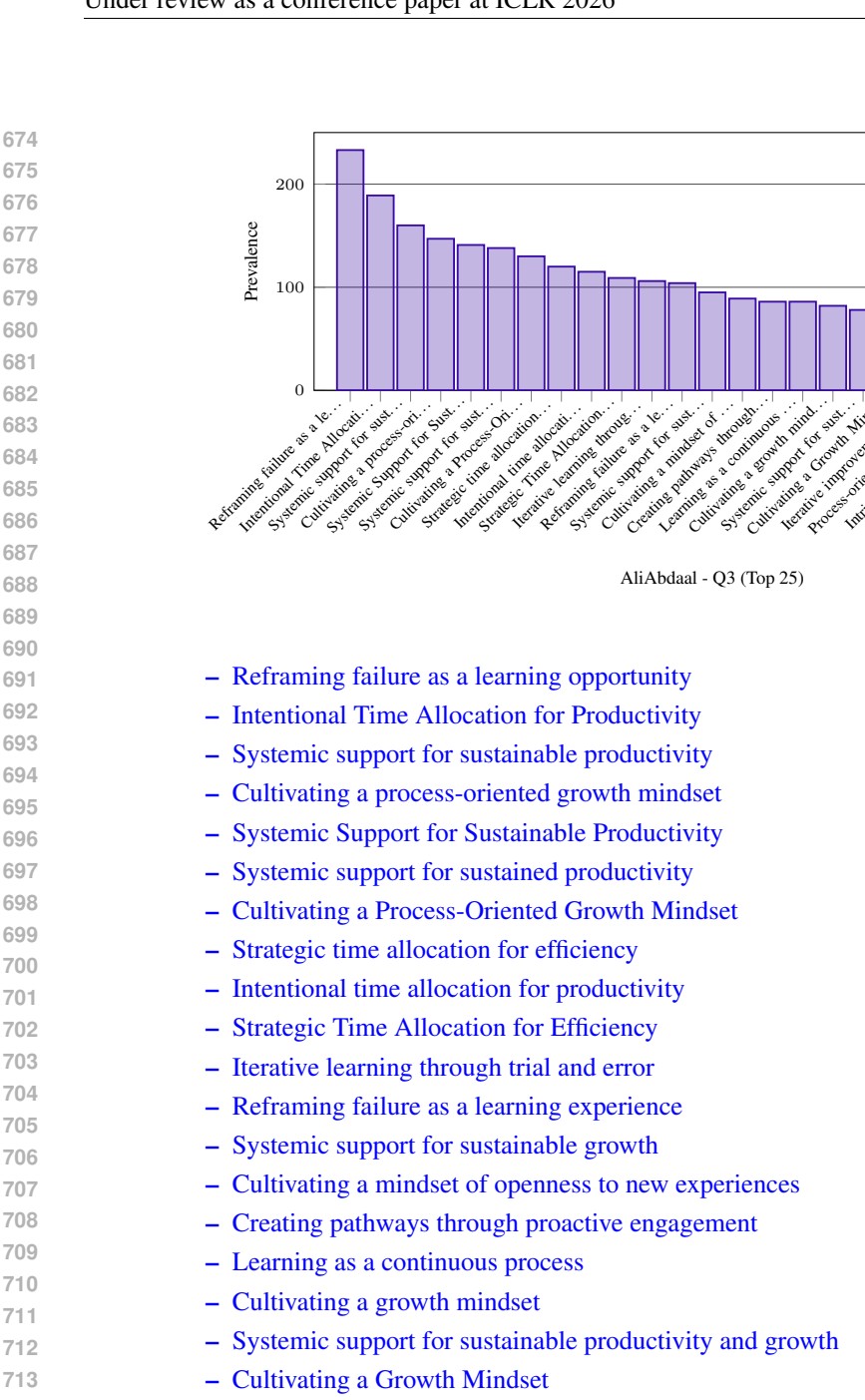

AliAbdaal - Q3 (Top 25)

- – Reframing failure as a learning opportunity
- – Intentional Time Allocation for Productivity
- – Systemic support for sustainable productivity
- – Cultivating a process-oriented growth mindset
- – Systemic Support for Sustainable Productivity
- – Systemic support for sustained productivity
- – Cultivating a Process-Oriented Growth Mindset
- – Strategic time allocation for efficiency
- – Intentional time allocation for productivity
- – Strategic Time Allocation for Efficiency
- – Iterative learning through trial and error
- – Reframing failure as a learning experience
- – Systemic support for sustainable growth
- – Cultivating a mindset of openness to new experiences
- – Creating pathways through proactive engagement
- – Learning as a continuous process
- – Cultivating a growth mindset
- – Systemic support for sustainable productivity and growth
- – Cultivating a Growth Mindset
- – Iterative improvement through feedback loops
- – Process-oriented growth mindset
- – Intrinsic motivation as core driver
- – Process-Oriented Growth Mindset
- – Systemic Support for Sustained Productivity
- – Enjoyment as Intrinsic Motivator

- **Q4:** In what ways does the speaker integrate or reference external sources (books, studies, tools), and what does this reveal about their epistemic stance? This question examines how external information is woven into the narrative—for example: as inspiration, validation, critique, or synthesis—thereby revealing whether the speaker leans toward evidence-based reasoning, personal interpretation, or community endorsement. By comparing integration patterns across the dataset, an underlying communicative strategy or epistemic stance can be inferred.

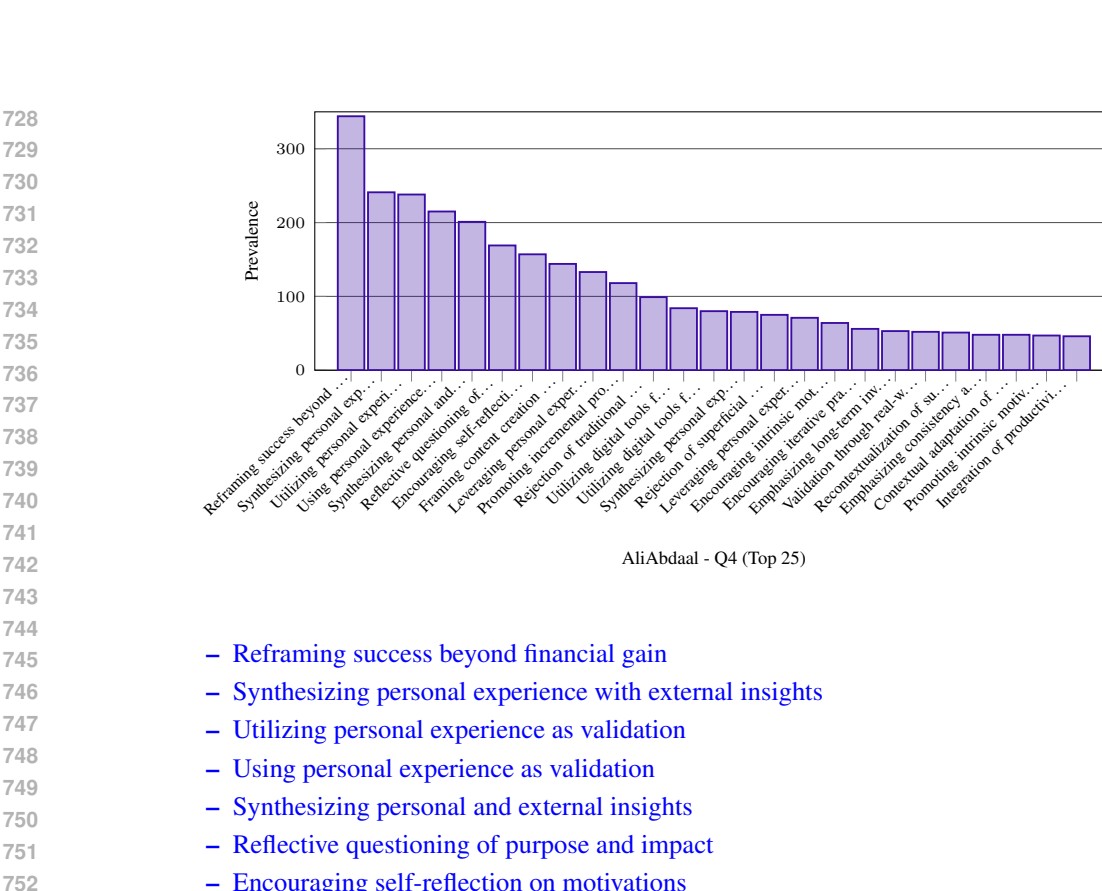

AliAbdaal - Q4 (Top 25)

- – Reframing success beyond financial gain
- – Synthesizing personal experience with external insights
- – Utilizing personal experience as validation
- – Using personal experience as validation
- – Synthesizing personal and external insights
- – Reflective questioning of purpose and impact
- – Encouraging self-reflection on motivations
- – Framing content creation as a long-term commitment
- – Leveraging personal experience as validation
- – Promoting incremental progress over perfectionism
- – Rejection of traditional productivity metrics
- – Utilizing digital tools for knowledge access
- – Utilizing digital tools for productivity optimization
- – Synthesizing personal experience and external insights
- – Rejection of superficial productivity metrics
- – Leveraging personal experience for validation
- – Encouraging intrinsic motivation over extrinsic rewards
- – Encouraging iterative practice for skill development
- – Emphasizing long-term investment in content creation
- – Validation through real-world application
- – Recontextualization of success beyond financial gain
- – Emphasizing consistency as foundational to skill development
- – Contextual adaptation of tools and techniques
- – Promoting intrinsic motivation over extrinsic rewards
- – Integration of productivity tools for workflow efficiency

- **Q5:** What recurring metaphorical or thematic language does the speaker use to communicate abstract concepts, and what conceptual schema emerges from those metaphors? This question encourages examination of language for consistent metaphoric or thematic patterns—such as "habits as bricks," "mind as a garden," or "productivity as flow"—and asks: "What cognitive frames or schemas do these metaphors evoke?" Rather than listing individual metaphors, the focus is on identifying broader conceptual categories they construct, such as "growth as cultivation" or "momentum as river," thereby revealing the speaker's underlying worldview or interpretive framework.

AliAbdaal - Q5 (Top 25)

- Learning as Iterative Process
- Intentional Time Allocation for Productivity
- Learning as Continuous Process
- Digital Tools as Cognitive Extensions
- Intentional time allocation for productivity
- Intentional Time Allocation
- Skill Development as Incremental Process
- Learning Through Iterative Practice
- Learning as continuous process
- Cognitive Offloading via Digital Tools
- Digital tools as cognitive extensions
- Learning as iterative process
- Productivity as Structured Systems
- Productivity as Sustainable Practice
- Growth through iterative practice
- Cognitive offloading via digital tools
- Productivity as Sustainable Practice through Focus
- Intentional Time Investment
- Content Creation as Iterative Process
- Intentional vs. Unintentional Time Use
- Time Allocation as Resource Management
- Time Management as Strategic Allocation
- Growth Through Iterative Practice
- Learning as Continuous Journey
- Enjoyment as Intrinsic Motivation

### A.6.4 MAS FAILURES DATASET QUESTIONS:

- **Q1:** Why does the multi-agent system fail to complete the task?

### A.6.5 MATH FAILURES DATASET QUESTIONS:

- **Q1:** Why does the proposed solution to this math problem fail, or what mistakes cause the incorrect result?

### A.6.6 Y COMBINATOR ACCEPTED APPLICATION DATASET QUESTIONS:

- **Q1:** What narrative strategies do YC-accepted applicants employ in 60-second pitch videos to establish founder credibility and execution capability, articulate and validate market problems through personal narratives and data, present differentiated solutions and competitive positioning, demonstrate market traction, define target markets and scalability, engage YC directly as audience, and structure persuasive pitches through opening/closing frames, simplification techniques, and team coordination?

### A.6.7 VIBE CODING DATASET QUESTIONS:

- **Q1:** How do programmers experience and make sense of vibe coding, including the workflows, iterative practices, challenges, and strategies they use to manage AI-generated code, integrate AI into development, and maintain control over project outcomes?

### A.7 TECHNICAL DETAILS

The repository contains six main areas:

- `LOGOS/` – Multi-iteration schema-induction and QA pipelines.
- `Lloom/` – A Lloom runner wired to local/remote vLLM backends.
- `GraphRAG/` – A customized GraphRAG runner for codebook generation.
- `LightRAG/` – A customized LightRAG runner for codebook generation.
- `HICode/` – A HICode runner with inductive coding and hierarchical clustring.
- `Thematic-LM/` – A ThematicLM reimplementation wired to local/remote vLLM backends.

### A.7.1 LOGOS

The LOGOS system is built as a multi-iteration schema-induction pipeline that integrates deterministic LLM-based coding with structural refinement across successive rounds. Default operation uses two iterations, conservative chunking, Qwen3-32B model for chat completions and Qwen3-embed-0.6b for embeddings, while enforcing deterministic text generation with zero temperature, bounded token outputs, and context windows truncated to a fixed budget. Prior to execution, the pipeline resets any intermediate artifacts to ensure reproducible runs.

At its core, the system orchestrates a multi-round induction loop that can expand to several iterations when needed. The first pass constructs a chunked corpus with controlled overlap, deterministic prompt formats, and embedding generation executed in parallel. Parameters such as concurrency of 32, overlap ratios at 200, and randomness seed 42 are fixed to ensure stable corpus formation. Later iterations reuse a mixture of previously computed embeddings and code assignments alongside newly generated ones. The system prioritizes reusing past representations for stability, while introducing embeddings of new candidate codes only when needed to cover semantic aspects of the chunk that were not previously captured. Each iteration refines—rather than rebuilds—the semantic graph: high-concurrency LLM selection updates or rescrores local codes, and a fallback heuristic

leverages graph connectivity and mergeability scores to select the strongest labels. Iterations continue until reaching the configured limit or until the graph stabilizes.

Memory-aware safeguards automatically tune chunk sizes, concurrency, and batching based on available RAM, ensuring that LOGOS can scale from resource-constrained machines to high-memory environments while maintaining throughput. The underlying corpus builder uses similarity thresholds, batched embedding inference, and deterministic generation settings to produce structured chunks labeled with source metadata, hierarchical depth, and tag information. When semantic similarity computation fails or returns degenerate vectors, fixed-dimension zero embeddings serve as a robust fallback.

Prompting strategies follow strict formatting rules, requiring pure JSON and mid-length textual codes, with separate modes for flat or hierarchical tag generation. During refinement rounds, the system applies conservative decoding, retry logic, token budget constraints, and downsampled candidate selection so that label synthesis remains focused and efficient even on large datasets. After completing the final iteration, LOGOS consolidates all outputs into a topologically ordered representation of the induced schema, along with summary statistics and, when the structure allows, a hierarchical tree describing the inferred semantic organization.

### A.7.2 LLOOM

LLOOM[5] is a relatively simple framework which only contain few core operators—Distill for text summarization, Cluster for semantic grouping, Synthesize for concept generation. We adpot the iteration of LOGOS for LLOOM pipeline, dropping the qouta mechanism in distill operation in LLOOM. Thus, the implementation details for LLOOM are the same to LOGOS on opencoding(Distilling), embedding and clustering(Clustering), and high-level code generation(Synthesizing).

### A.7.3 GRAPHRAG

We used the open source GraphRAG repo[6], and configured to define the model choices, concurrency, and retrieval behavior. In our experiment, we use the Qwen3-32B model for chat and Qwen3-embed-0.6b for embeddings, with high concurrency settings (chat up to 32, embeddings 16 async) and generous retry limits. Documents are chunked at 600 tokens with an overlap of 50. Query and report generation are constrained by token limits (up to 12,000 tokens), and generated community reports are capped at 8,000 tokens. We modify the report generator prompt to directly ask LLM to generate 100 item codebook. The query runner processes questions from a CSV file, writes results to another, includes a drift-focused search scope, and enforces delays and retry logic.

### A.7.4 LIGHTRAG

We follows the LightRAG repo[7] to run the baseline. It operates with large-batch processing defaults (1,000 items per batch) and adaptive behavior for big datasets. We chunks documents into 500-token segments with 50-token overlap and retrieves top-k=20 results per query. The pipeline use Qwen3-embed-0.6B as embedding model, asynchronous embedding concurrency of 16, and Qwen3-32B as chat LLM with concurrency of 32, context windows up to 16,000 tokens. Query fallback logic progressively reduces top-k and max-token limits, and applies delays between attempts. Its environment template supports caching and high token ceilings, while the storage layer connects to Neo4j, Redis, Postgres HNSW indexes, and Qdrant via specified performance-tuned settings.

---

[5]https://stanfordhci.github.io/lloom/
[6]https://github.com/microsoft/graphrag
[7]https://github.com/HKUDS/LightRAG

### A.7.5 HICODE

Following HICode repo[8], we implement the inductive coding and hierarchical clustering pipeline using `Qwen/Qwen3-32B` (non-thinking mode) for both label generation and clustering. Label generation operates with temperature = 0.0, JSON-formatted deterministic outputs, and max tokens = 8192, with up to 3 retries using exponential backoff. During clustering, all labels are first deduplicated and partitioned into batches of 100, which are processed asynchronously with a semaphore limit of 32 concurrent requests and a 600-second timeout per call. Hierarchical clustering proceeds for 3 iterations, where each iteration applies LLM-based cluster synthesis independently to each batch, followed by deterministic merging of cluster dictionaries. Subsequent iterations cluster the theme labels produced in the previous round, yielding progressively more abstract themes. All experiments use the same model and hyperparameters across datasets unless otherwise specified.

We used the final codebook + theme produced from the train set from **the final iteration** as their final codebook to be applied on test set.

### A.7.6 THEMATIC-LM

Since the original paper did not make their codebase publicly available, we attempted to reimplement a similar sequential batched LLM-based aggregation-and-review mechanism following the paper articulation. Because the textual segments in our dataset (e.g. in MAS, Ali dataset) are typically significantly longer than the Reddit post dataset used in the original paper, and the text span corresponding to a single code is much longer (e.g. the code 'inter-agent communication failure' on MAS), we decided to remove the quotation design from the paper, leaving only the codes. We used only 1 persona but producing 15 codes for each datapoint (with prompting the open coding agent to think from diverse perspectives) for fair comparison with other methods.

we implement a batched sequential aggregation pipeline using `Qwen/Qwen3-32B` (non-thinking mode) for code aggregation and `qwen3-embed-0.6b` for semantic similarity matching. The system processes raw open codes in batches of 100 datapoints, where each batch undergoes LLM-based code aggregation with temperature=0.0 for deterministic outputs, followed by a reviewer agent that merges new labels with the global codebook using cosine similarity threshold of 0.65. All API requests are processed asynchronously with a semaphore limit of 16 concurrent requests and 120-second timeout per call, utilizing multiple chat endpoints for load balancing. The pipeline employs cached embeddings with batch size 256 to optimize performance, and implements three parallel theme coders that generate higher-level thematic groupings from the final codebook, which are then synthesized through a theme aggregator. JSON outputs are strictly enforced with think blocks stripped, and the system supports both incremental batch processing and full-dataset processing with configurable code limits for rapid prototyping.

We used the final codebook + theme produced from the train set as their final codebook to be applied on test set.

### A.8 ETHICAL STATEMENTS

We used LLM (GPT, Gemini) to improve the sentence and grammar for writing the abstract, introduction, experiments, and conclusion sections of the paper. All text are mannually written and LLM only helps with word choice and grammar. We also used LLMs to generate python codes to produce the figures in this paper. We mannually checked the numbers to ensure they are entered correctly.

---

[8]https://github.com/mianzg/HICode

