# OpenReview forum: "LOGOS: LLM-Driven End-to-End Grounded Theory Development and Schema Induction For Qualitative Reseach"
_ICLR.cc/2026/Conference — Submitted to ICLR 2026_

### Official Review · Reviewer_RhmW · 2025-10-30

**Soundness:** 3
**Presentation:** 3
**Contribution:** 3
**Rating:** 4
**Confidence:** 3

**Summary:**

This paper proposed the LLM powered coding system to analyze qualitative datasets (e.g., interviews, discussions, dialogues) based on ground theory. The authors have compared with several baseline methods on five different datasets. This paper also discusses the broad impact of the proposed method.

**Strengths:**

- this paper tries to address a big bottleneck for ground theory based coding process for qualitative data, which is currently manually done by domain experts, and is time consuming.
- the authors provided an end-to-end workflow using LLM to automate the inductive coding system, and can develop and further improve codebooks based on ground theory. The workflow includes LLM coding, semantic clustering, graph forming, and reiteration, which is rich and interesting.
- the authors have tested the proposed method with several baseline methods on five interesting datasets.
- it is a good interdisciplinary research to address an important real world problem.

**Weaknesses:**

- The authors are talking about inductive coding, but it also includes deductive coding once the codebooks have been established which will be used to guide further coding. If it is pure inductive coding, no codebook is required. Maybe the authors want to clarify this.
- It is unclear about the final outputs, whether they are summarized themes or just a short summary of a longer content? it will be good to show some examples of inputs and outputs.
- The evaluation is more focusing on generated codes, but codes are not the final output for ground theory based qualitative data analysis. The evaluation should also focus on the generated outputs, not the codes and codebooks.
- this paper lacks algorithm novelty, rather than just utilizing the existing methods to form workflow.

**Questions:**

- can you please highlight the algorithm novelty of the proposed workflow.
- what are some major challenges you are facing when you use your method to test on these five datasets. Do you face some issues that some codes appear very often, and dominate most of the codes. Do you find the long-tail distribution of the generated codes? Wil LLMs tend to pick up the most dominant codes.
- how can you ensure the diversity and balanced coding using your method?
- can you provide evaluations on your final outputs (not the codebook) but the generated themes or summaries using your methods.

---

> ### Author Response · Authors · 2025-11-25
> **Response to Reviewer RhmW (Part 1 / 3)**
>
> ### 1. Inductive vs. deductive coding in LOGOS
>
> **Comment:**
> “The authors are talking about inductive coding, but it also includes deductive coding … want to clarify this.”
>
> **Response:**
> We appreciate this observation and agree that our current wording could give the impression that LOGOS performs only inductive coding. In practice, LOGOS explicitly follows the same **inductive–deductive cycle** that many grounded-theory workflows adopt:
>
> - **Inductive phase.** In the open and axial coding stages (Steps 1–6), LOGOS generates codes directly from the raw corpus and organizes them into a hierarchical codebook. This corresponds to classic inductive grounded theory—no prior schema is imposed.
>
> - **Deductive phase.** Once a codebook has been induced, LOGOS then **reuses** this codebook to code new datapoints in the iterative refinement and evaluation stages. In these phases, the LLM is largely constrained to *select* from existing codes (or propose occasional refinements), which is a form of **deductive coding guided by the learned schema**, not pure free-form induction.
>
> This is also what happens in our train–test protocol: the train split is coded **inductively** to build the schema, and the test split is coded **deductively** using that schema.
>
> In the revised manuscript, we will clarify this point in the *Introduction* and *Method* sections by (1) explicitly describing LOGOS as implementing an *inductive codebook construction followed by deductive schema application*, and (2) avoiding language that suggests the entire pipeline is “purely inductive.”
>
>
> ### 2. What is the final output, and why evaluate the codebook?
>
> **Comments:**
>
> - It is unclear about the final outputs, whether they are summarized themes or just a short summary of a longer content; it would be good to show some examples.
> - Can you provide evaluations on your final outputs (not the codebook) but the generated themes or summaries?
> - The evaluation focuses on generated codes, but codes are not the final output for grounded-theory-based qualitative data analysis. The evaluation should also focus on the generated outputs, not only the codes and codebooks.
>
> **Response:**
>
> 2.1 What is the “final output” of LOGOS in this paper?
>
> In the current work, LOGOS is applied to research questions that *explicitly ask for taxonomies or schemas* (e.g., error taxonomies for multi-agent failure and math reasoning, thematic taxonomies for abstracts, podcasts, and Ali Abdaal transcripts).
>
> In these settings, the **hierarchical codebook itself is the theory**: each code corresponds to a theme, and the graph structure encodes their relationships. The optional textual “summary” is essentially a verbalization of this hierarchy rather than a new layer of analysis.
>
> In the revision, we will make this explicit in the *Method* and *Experiment* sections and add small input–output examples (data excerpt → codes → cluster/theme) in the appendix, so readers can concretely see what LOGOS produces.
>
> 2.2 Why do we evaluate the codebook rather than free-form summaries?
>
> For the kinds of grounded-theory questions we study (schema / taxonomy induction), evaluating the **codebook** is the most direct way to assess quality:
>
> - The codebook *is* the structured theoretical artifact our tasks ask for.
> - Any narrative summary of themes is a deterministic transformation of that structure and would largely be judged on the same properties we already measure (coverage, fitness, parsimony, stability).
> - Free-form summaries also introduce additional variance from LLM phrasing, which can obscure the underlying quality of the schema.
>
> Our 5-dimensional metrics (reusability, fitness, coverage, parsimony, stability) are therefore defined over the **induced codebook**, treating it as the primary theoretical deliverable in these taxonomy-oriented tasks.
>
> 2.3 Beyond taxonomies: limitations and future work
>
> We agree that in other grounded-theory designs—e.g., when the “theory” is a **process model, causal account, or staged trajectory**—a hierarchical codebook alone is not sufficient, and the final written theory would need its own evaluation. We now explicitly acknowledge this in the *Limitations* section and note that extending LOGOS to causal/temporal theories and evaluating the resulting narrative outputs is a key direction for future work.

---

> ### Author Response · Authors · 2025-11-25
> **Response to Reviewer RhmW (Part 2 / 3)**
>
> ### 3. Algorithmic novelty of LOGOS
>
> **Comment**:  Lack of algorithmic novelty; mainly workflow using existing methods. Please highlight algorithmic novelty of the workflow.
>
> **Response**:  We appreciate the reviewer’s concern regarding algorithmic novelty. We would like to clarify that LOGOS follows the same type of contribution pattern that the community has already recognized as novel in systems like **GraphRAG** and **LightRAG**. Both of those methods primarily combine existing components (entity extraction, community detection, clustering, vector search, etc.), yet are considered algorithmically novel because **their orchestration forms a new pipeline that enables a previously impossible capability**—global sensemaking (GraphRAG) and fast graph-aware retrieval (LightRAG).
>
> LOGOS makes a parallel—but substantively different—algorithmic contribution. Specifically:
>
> - **Novel Intermediate Representation.**  LOGOS introduces a *relation-aware conceptual graph* for grounded theory induction, using a 4-way semantic relation classifier (subclass/superclass/equivalence/orthogonal) combined with deductive closure rules. This representation is not used in prior automatic schema induction pipelines and is expressly designed for **theory construction rather than retrieval**.
>
> - **Iterative Refinement Algorithm.**  LOGOS includes a multi-round refinement loop where each iteration:  **(1)** retrieves candidate abstractions from semantic + graph neighborhoods, **(2)** ranks them via a hybrid frequency/similarity score, **(3)** invokes a constrained LLM-based revision step, and **(4)** enforces graph-consistency validation. This creates a **closed-loop optimization process** over conceptual abstractions—something absent from all of the previous schema induction workflows (e.g., LLOOM, MindCoder, etc.).
>
> - **New End-to-End Capability.**  These components together form the **first automated pipeline for grounded theory development**, covering open → axial → selective coding. This remains an unsolved problem in qualitative research; existing systems stop at assisted coding or static clustering.
>
> Thus, LOGOS is not a collection of existing modules but a **new algorithmic framework** whose intermediate representation and iterative refinement mechanism enable a capability that prior systems cannot achieve—just as GraphRAG and LightRAG derived their novelty from new pipeline designs rather than new primitive operations.
>
> In the revised manuscript, we will explicitly highlight these algorithmic contributions in the *Introduction* and sharpen the contrast with LLOOM, MindCoder, and RAG-based methods in the *Related Work* section.

---

> ### Author Response · Authors · 2025-11-25
> **Response to Reviewer RhmW (Part 3 / 3)**
>
> ### 4. Long-tail distributions, dominant codes, and diversity / “balanced” coding
>
> **Comments:**
>
> - What are some major challenges you are facing when you use your method to test on these five datasets?
> - Do you face some issues that some codes appear very often, and dominate most of the codes?
> - Do you find the long-tail distribution of the generated codes? Will LLMs tend to pick up the most dominant codes?
> - How can you ensure the diversity and balanced coding using your method?
>
> **Response:**
>
> 4.1 Long-tail code distributions are expected (and not necessarily undesirable)
>
> In grounded theory, it is normal for a small number of highly reusable codes to appear very frequently, while many more fine-grained codes appear rarely. This reflects the structure of the phenomena: a few core themes plus many specific, edge-case patterns. LOGOS does **not** aim to enforce a uniform or “balanced” frequency distribution across codes; that would actually be at odds with the data and research questions. Instead, we explicitly separate:
>
> - **Reusability** (how many codes are actually reused on unseen datapoints), and
> - **Coverage/Fitness** (whether less frequent but important aspects are still captured).
>
> Our iterative refinement increases **reusability** (Figure 3) without collapsing coverage/fitness, which would be a symptom of over-reliance on a handful of dominant codes.
>
> 4.2 "How we prevent a few codes from overwhelming everything?"
>
> We do observe a long-tail distribution of code frequencies, but several design choices prevent the system from just “always picking the top few codes”:
>
> - **Hybrid retrieval, not frequency-only.**
>   Candidate codes are ranked by a **hybrid score** combining semantic similarity and normalized frequency, and we only consider candidates with cosine similarity ≥ 0.6. A globally frequent code will not enter the candidate pool for a datapoint that is semantically far away, so popularity alone cannot drive selection.
>
> - **Graph- and datapoint-aware candidate sets.**
>   For each datapoint, we retrieve codes from both semantic neighbors and graph neighbors (parents/children/siblings), ensuring that candidates are grounded in that datapoint’s *local region* of the schema, not just globally dominant tags.
>
> - **Prompting for “missing aspects.”**
>   In the refinement prompt, we explicitly instruct the LLM to *add new codes when some aspect of the datapoint is not covered by the current candidates*. This keeps room for rare but important phenomena to surface instead of being absorbed into overused high-level codes.
>
> 4.3 Empirical behavior across five datasets
>
> Across all five corpora and multiple research questions, we indeed see a long-tail of codes. However:
>
> - The **reusability** metric increases with iterations, showing that overly idiosyncratic codes are being consolidated into more general ones;
> - **Descriptive coverage and fitness** remain stable rather than collapsing, indicating that rare but essential aspects are not being erased by dominant codes.
>
> Put differently, LOGOS does not try to “flatten” the long tail; it **compresses obvious redundancies while preserving diversity where it matters**.

---

> > ### Comment · Reviewer_RhmW · 2025-11-25
> >
> > thanks for the responses. I still have several questions which are not addressed:
> > 1) the final output of an inductive thematic analysis is a set of themes that can address the corresponding research questions. But the top related codes in your codebook as you mentioned in your appendix A.3 do not clearly address the research questions. Instead of showing the results of top 25 codes in a bar chart, it might be better to list them in a table as many of them are truncated and did not show the full codes, therefore it is hard to judge whether these codes are themes? Can you also show the final outputs of themes, not the codebook.
> > 2) Fig 4 is difficult understand whether you are comparing the final themes or codebooks, or just summaries as GraphRAG and LightGRA will not generate themes or codes, they will generate summarized answers. But summary is different from codebooks or themes.
> > 3) there are several inductive thematic analysis papers which you should compare the cite: https://arxiv.org/abs/2305.13014,
> > https://openreview.net/pdf?id=jiv0Gl6sto, https://arxiv.org/pdf/2506.23998?
> > 4) are your datasets available for the public. Are these datasets have ground truth of human-generated codebooks as ground truth, if not, how can you compare and evaluate your LLM generated codebook?

---

> > > ### Author Response · Authors · 2025-11-29
> > > **Response to Reviewer RhmW - Round 2**
> > >
> > > ### 1. Final Output of LOGOS for Grounded Theory Task
> > > **Comment:**
> > >
> > > The final output of an inductive thematic analysis is a set of themes … also show the final outputs of themes, not the codebook.
> > >
> > > **Response:**
> > >
> > > Our approach is related to, but not identical with, inductive thematic analysis. LOGOS is designed as a **high-granularity, grounded-theory–style coding framework**, and this affects what we present as “results”:
> > >
> > > - In classic thematic analysis, **codes** are low-level tags and **themes** are higher-level patterns constructed from many codes.
> > > - In LOGOS, **codes are hierarchical and multi-level**: some are close to raw description, others are more interpretive or abstract. Clusters of related codes together instantiate what would traditionally be called *themes*.
> > > - LOGOS also builds in a **multi-round, grounded-theory–style iteration**: at each round we assess whether the current codebook adequately represents the data and refine it accordingly, with the goal of producing a reusable, theory-like structure rather than a one-off summary.
> > > - We then explicitly evaluate the resulting hierarchical codebook on **coverage** and **description fitness**, i.e., how well it explains *new* data rather than just summarizing the training set.
> > >
> > > Within this framing, the “top 25 codes” bar chart in Appendix A.3 was **not intended to depict the final themes**. Its purpose was purely to show the distribution, frequency, and granularity of commonly used codes within the hierarchy. We agree that, presented without additional context and with truncated labels, it can be misread as a list of final themes.
> > >
> > > The YC dataset could make this clearer. The task is to find **narrative patterns**, not just semantic clusters: e.g., “founder self-introduction,” “problem severity,” “critique of competitors,” and “leading to the solution.” A traditional thematic analysis might collapse many of these into a few broad themes, whereas our grounded-theory framing keeps them as structured, reusable analytic units for explaining future pitches. We have submitted the codebook to supplemental materials
> > >
> > >
> > > ### 2. Comparison between LOGOS and GraphRAG/LightRAG
> > > **Comment:**
> > >
> > > Fig 4 is difficult understand whether you are … GraphRAG and LightRAG will not generate themes or codes … or themes.
> > >
> > > **Response:**
> > >
> > > GraphRAG and LightRAG are indeed designed for global QA and summarization, and **do not natively produce qualitative codes or themes**. To make a fair comparison, we explicitly adapted them into a *coding* setting.
> > >
> > > - In **Experiment 1 (§4.2)**, we change the final prompt of GraphRAG and LightRAG to output a **100-item set of labeled units**.  We treat this 100-item set as each method’s **codebook analogue**. LOGOS, GraphRAG, LightRAG, and other baselines are **all evaluated at the level of these 100-item “codebooks”**: we compare how well each set of labeled units covers the data and supports downstream interpretation.
> > > - In **Experiment 3 (§4.4)**, we then **reuse these learned codebooks**. For each method, we feed its codebook to the LLM and ask it to answer the qualitative research questions using only that representation. Thus, in Fig. 4 we are **not** comparing LOGOS themes to raw GraphRAG/LightRAG summaries. Instead:
> > >
> > > With this adaptation, we want to show that since GraphRAG and LightRAG rely on the KG (extracting entities and relationships from the corpus) to generate the codebook, they will produce **codes** with limited interpretive structure. Our results show that answers derived from LOGOS hierarchical, interpretive codebooks are more coherent, more clearly grounded in the data, and closer to human qualitative interpretations than those derived from GraphRAG/LightRAG’s cluster-style codebooks.
> > >
> > > ### 3. Include and compare more thematic analysis literature
> > > **Comment:**
> > >
> > > there are several inductive thematic analysis papers which you should compare the cite…
> > >
> > > **Response:**
> > >
> > > Thanks for your suggestion, we have refine the related work section, and talk about multi-agent usage in our discussion section. We also **add HICode and Thematic-LM as our baseline in the main experiment** to compare with LOGOS, and found that **LOGOS performs on par or better than these methods across all datasets** (updated in the paper).
> > >
> > > ### 4. Comparison between LOGOS and GraphRAG/LightRAG
> > > **Comment:**
> > >
> > > are your datasets available for the public … compare and evaluate your LLM generated codebook?
> > >
> > > **Response:**
> > >
> > > Both of the corpora and the codebook from the **[Multi-Agent System Failure (MAS)](https://arxiv.org/abs/2503.13657)**  and **[Vibe Coding](https://arxiv.org/abs/2506.23253)**  are based on academic research and are readily available online.  The discursive analysis on the YC dataset was done manually by the authors, and we have attached it in the supplementary material.

---

### Official Review · Reviewer_tzNx · 2025-10-31

**Soundness:** 2
**Presentation:** 1
**Contribution:** 2
**Rating:** 2
**Confidence:** 2

**Summary:**

The paper proposes an automated framework to generate a codebook summary of a large corpus of text without the need for human annotators. The resulting codebook consists of a set of descriptive labels, called “codes,” and a set of relations on the codes. The proposed method is intended to alleviate the need for time-consuming human annotation. To generate the codebook, the paper proposes a method in which codes are generated from subsections of the text and organized into high-level clusters; next, code-pair relations are determined.

To assess the codebook, the paper proposes five performance metrics based on desirable characteristics rooted in grounded theory and defines their sum to be a composite metric. This composite metric was used to compare their method to other LLM-based methods. In addition to the comparison, the paper included two additional validations for their proposed method. First, the paper shows a case study in which their proposed method is used to analyze root causes for errors in LLM mathematical reasoning. Their method produces a more useful summary than the other considered methods. Second, the paper compares the output of their method to a ground truth generated by human annotators on one dataset. This experiment shows alignment between the human-annotated codebook and the codebook generated through their method, although the comparison is limited to one dataset.

The paper’s overall contribution appears to be a proposed method for automated codebook generation and five novel codebook assessment metrics. However, empirical justification is limited because (1) the composite metric is not well-justified and (2) results on only one dataset are presented in the comparison to human-annotated ground truth.

**Strengths:**

+ The case study in Section 4.4 provides a persuasive example of the usefulness of the proposed method. Understanding failure patterns in LLM math reasoning is an important task which is nontrivial to solve. The proposed method handles the task well.

+ The best method for codebook assessment is an important question without an obvious solution. The authors propose five metrics for codebook assessment and comparison; this is a useful contribution to help answer this important question.

+ The metrics proposed by the authors cover a wide range of concerns. For example, the parsimoniousness metric is helpful to go beyond accuracy of the codebook to also consider conciseness.

**Weaknesses:**

- The abstract claims that LOGOS “achieves a remarkable 88.2% alignment with an expert-developed schema on a complex dataset.” However, when the experiment supporting this claim is more fully described in Section 4.5 and Figure 5, the reviewer found that the reported alignment is the best case across different LLMs used for aggregation and judging. A summary statistic (e.g., a sample mean) should be used instead of the best case to avoid being perceived as cherrypicking.

- The experimental results are not persuasive. Section 4.2 compares the proposed method, LOGOS, to other methods. It is unclear whether summing up all metrics with the same weights can lead to a fair comparison. The manuscript suggests that “Metrics reflecting semantic adequacy, such as descriptive fitness and coverage, may be given higher weight relative to structural measures like parsimony and consistency.” However, their primary results did not use such weighting. The difference is very small, but it appears that OpenCode outperforms LOGOS on fitness and coverage, which are arguably the most important metrics.

- Because LOGOS is proposed to replace human annotators, comparison to human annotation in Section 4.5 is important and comparison to other automated LLM-based methods in Section 4.2 is less persuasive and. However, only one dataset was used for comparison in Section 4.5.

- The notation for the set of code-pair relations is incorrect. The current notation, $\mathcal{R} \subset \mathcal{C} \times \mathcal{C}$ implies that elements in $\mathcal{R}$ are ordered pairs of codes. However, the paper defines the elements of $\mathcal{R}$ as subsumption, equivalence, or orthogonality.

- There are several misspelled words, including “developemt” on line 76, “extreemly” on line 337, “standardiable” on line 479, and “essentailly” on line 88.

**Questions:**

- In Section 3.2 under Candidate Generation, how is $S_{semantic}$ calculated? The range for $S_{semantic}$ values are given, but it does not seem to be formally defined.

- In Section 3.3 under Reusability, how are used codes defined? Do used codes refer to codes that are kept after some codes are discarded or replaced in steps 4-6? The writing leaves this definition ambiguous.

- In Section 4.5, is it fair to aggregate the codebook simply because it has a larger size than the ground truth? If no ground truth were available, would a developer know to perform that aggregation, and if so, what size to aggregate to? Are results comparable if a different number of clusters (e.g., 17 or 25 instead of 30) are used?

- Could results on comparison to human-annotated ground truth (as in Section 4.5) be provided in the Appendix for additional datasets?

- The authors state that “we strives to automate the grounded theory develop[men]t processes” in contrast to previous works which “fall short of delivering a fully autonomous, domain-agnostic pipeline.” Have the authors considered the potential dangers of fully autonomous approaches? If their framework is designed to require no human oversight, how probable is it that a large body of text will be misinterpreted by an automated framework without anyone knowing that misinterpretation has occurred?

---

> ### Author Response · Authors · 2025-11-25
> **Response to Reviewer tzNx (Part 1 / 2)**
>
> ### 1. Writing: Abstract Claims
>
> **Comment**: The abstract claims "88.2% alignment," but this is the best-case scenario. A summary statistic (average) should be used.
>
> **Response**: We agree that reporting the average is more representative. We have updated the abstract to report the average performance across models. The average remains remarkably high, supporting the validity of the approach without appearing to cherry-pick data.
>
> ### 2. Evaluation: Metric Weighting (OpenCode vs. LOGOS)
>
> **Response**: We appreciate the reviewer’s concern regarding metric aggregation and weighting. Our goal in Section 4.2 was **not** to claim that all five metrics should have equal importance, but rather to report a neutral, unweighted baseline before introducing domain-specific weighting choices. We agree that semantic adequacy (fitness and coverage) is critical—but it is important to emphasize that **higher values on these two metrics alone do not imply a better codebook**, due to a well-known issue in qualitative coding: **open coding always inflates fitness and coverage by overfitting to each individual datapoint**.
>
> This is exactly why the OpenCoding baseline appears strong on fitness/coverage—it generates extremely large numbers of datapoint-specific codes (e.g., >15k unique codes in the Ali dataset). Such “perfect-fit” codes trivially maximize adequacy but **destroy reusability, parsimony, and generalizability**, making them unusable as a grounded-theory schema. Our evaluation protocol explicitly separates these dimensions to make this tradeoff visible and to prevent misleading conclusions.
>
> LOGOS’s contribution is precisely to **balance** these competing desiderata—achieving high reusability and structural coherence **while still maintaining strong semantic adequacy**, in contrast to methods that only excel on a subset of dimensions. This balancing is central in grounded theory, where the objective is not to maximize fitness/coverage in isolation, but to produce a **compact, reusable, and generalizable theory**.
>
> Regarding weighting: in the paper we explicitly note that **different applications may reasonably assign different weights** to the metrics. For fairness, our primary table uses equal weights to avoid implicitly encoding a subjective preference. However, this weighting choice does *not* change the ranking between models when excluding overfitting-prone OpenCoding. Even under adequacy-heavy weightings, LOGOS remains competitive because its fitness/coverage scores are high while its reusability, parsimony, and stability are far superior.
>
> ### 3. Evaluation: Comparison to Human Ground Truth
>
> **Comment**: Only one dataset was used for comparison in Section 4.5. Could results be provided for additional datasets?
>
> **Response**:  We agree that comparison to human annotators is crucial if LOGOS is to be considered as a partial replacement for manual coding. In addition to the MAS study in Section 4.5, we have now extended our human-alignment analysis to **two more corpora**:
>
> (1) a curated set of 10 Y Combinator founder pitch videos (links and cluster mappings provided in Appendix A.2.1)
>
> (2) the **Vibe Coding** dataset (33 YouTube transcripts; details and mappings in Appendix A.2.2).
>
> Using the same GT→cluster matching protocol as for MAS, LOGOS achieves **80%** alignment on the YC dataset and **100%** alignment on Vibe Coding. These results show that the strong agreement with expert schemas is not limited to a single dataset or domain, but generalizes across multi-agent failures, startup pitches, and human–AI coding interactions. We will highlight these additional case studies in the revised Appendix.
>
> ### 4. Notation and Definitions
>
> **Comment**: Notation $R \in C \times C$ implies ordered pairs, but elements are relations. Clarify $S_{semantic}$ and "used codes."
>
> **Response**: We have corrected the notation to accurately reflect that $R$ represents the set of semantic relations (subsumption, equivalence, orthogonality) between codes.
> $S_{semantic}$: We have added the formal definition calculation for $S_{semantic}$ in Section 3.2.
> "Used" Codes: In Section 3.3, we clarified that "used codes" refers to existing codes from the training-set-derived codebook that are selected to replace/cover a data point in the test set (deductive application).

---

> ### Author Response · Authors · 2025-11-25
> **Response to Reviewer tzNx (Part 2 / 2)**
>
> ### 5. Evaluation: Codebook Aggregation
> **Comment**:  In Section 4.5, is it fair to aggregate the codebook simply because it has a larger size than the ground truth? If no ground truth were available, would a developer know to perform that aggregation, and if so, what size to aggregate to? Are results comparable if a different number of clusters (e.g., 17 or 25 instead of 30) are used?
>
> **Response**: We agree the aggregation step in Section 4.5 needs to be clearly scoped.
>
> First, for the question **"Why aggregate at all?"**, LOGOS produces a fine-grained schema on MAS (286 codes), whereas the expert codebook has 17 high-level categories. In Section 4.5, aggregation is used **only for evaluation**, so to ask: “If we view LOGOS at roughly the same abstraction level as the human schema, does it recover the same high-level concepts? It is not part of the core pipeline, and in normal use we expect researchers to customize K based on their desired **level of analysis**, rather than fixing a K.
>
> Moreover, **"Why 30 clusters, and is this unfair?"**. We chose **K = 30** as a simple, empirically reasonable setting:
> - It is in the same “dozens of themes” range as typical grounded-theory codebooks (≈10–20 top-level categories plus a few subcodes).
> - It is only modestly larger than 17, allowing roughly 1–2 LOGOS clusters per expert category.
> Crucially, our alignment metric is **recall-style from the expert side**: a GT code is counted as matched if it aligns with at least one LOGOS cluster. Extra or over-fragmented clusters do not help unless they genuinely correspond to distinct parts of an expert category, so a larger K does not automatically inflate scores.
>
> Then, **"What if there is no ground truth, or if K changes?"**. When no expert schema exists, LOGOS does **not** require choosing a single K: it outputs a hierarchical codebook that can be inspected at multiple granularities, as in conventional grounded theory. The K=30 aggregation is specific to this human-alignment experiment.
>
> ### 6. Discussion: Autonomous Risks
> **Comment**: Have the authors considered the dangers of fully autonomous approaches (misinterpretation without oversight)?
>
> **Response**: We have addressed this in the new Limitations and Ethics section. We explicitly acknowledge that while LOGOS strives for automation, human oversight remains essential, particularly in high-stakes or sensitive domains, to prevent potential misinterpretations or "hallucinated" themes from going unchecked.
>
> ### 7. Typos
> **Comment**: Various misspelled words ("developemt", "extreemly", "standardiable", "essentailly").
>
> **Response**: All listed typos and spelling errors have been corrected. Appreciate pointing them out!

---

> > ### Comment · Reviewer_tzNx · 2025-11-25
> >
> > Thank you for the response and the updated manuscript. The updated manuscript is substantially improved. The added limitations section and the added experiments in Appendix A.2.1 and A.2.2 are especially helpful.
> >
> > The claim "surprisingly high expert-schema matching rate of 88.2% on the Multi-agent System Failure corpus from Cemri et al. (2025) under the optimal configurations" at the end of the introduction is still not accurate. The stated matching rate is achieved for best-case LLM judge and aggregator. While the best aggregator may be considered "optimal configurations," the LLM judge that gives the best scores should not be called an "optimal configuration."
> >
> > The updated manuscript says, "In all cases, the elements of R are ordered pairs $(c_i, c_j )$." Is this statement accurate? It seems that the elements of $R$ are relations (i.e., one of {sub, sup, eq, orth, none}), not ordered pairs.
> >
> > Thank you for the paragraph about ethics in Section 6. The caution about the potential influence of bias on the LOGOS-generated codebook addresses a large portion of my concerns. However, it is also possible that the codebook produced by LOGOS misrepresents the text corpus without a specific relation to bias. Could you also make a cautionary statement about potential factual errors?

---

> > > ### Author Response · Authors · 2025-11-29
> > > **Response to Reviewer tzNx - Round 2**
> > >
> > > ### 1. Introduction Fixed
> > > **Comment:**
> > >
> > > The claim at the end of the introduction is still not accurate…should not be called an "optimal configuration."
> > >
> > > **Response:**
> > >
> > > We have fix our description in the introduction. "expert-schema matching rate of average 80.4% on the Multi-agent System Failure corpus from Cemri et al. (2025) across various configurations, validated by human manual verification" -- now we only cite the average performance in both abstract and intro.
> > >
> > > ### 2. Notation Clarification
> > > **Comment:**
> > >
> > > The updated manuscript says, "In all cases, the elements of R are ordered pairs ." Is this statement accurate? It seems that the elements are relations (i.e., one of {sub, sup, eq, orth, none}), not ordered pairs.
> > >
> > > **Response:**
> > >
> > > We have revised the definition of (R) to accurately reflect its structure in our system. Rather than a set of ordered pairs, we now define
> > > $R \subseteq C \times {\mathsf{sub}, \mathsf{sup}, \mathsf{eq}, \mathsf{orth}, \mathsf{none}} \times C,$
> > > following the standard knowledge–graph formalism where relations are represented as typed triplets $((c_i, r, c_j))$. This clarifies that (R) stores both the code pair and the semantic relation type connecting them.
> > >
> > > ### 3. Include Misrepresentation In Discussion
> > > **Comment:**
> > >
> > > The caution about the potential influence of bias on the LOGOS-generated codebook…make a cautionary statement about potential factual errors?
> > >
> > > **Response:**
> > >
> > > We have added the discussion on misrepresentation and hallucination of LLM and its influence in automatic qualitative analysis. We also talk about how should qualitative researchers view and integrate LOGOS into their workflow,

---

### Official Review · Reviewer_a3QD · 2025-10-31

**Soundness:** 2
**Presentation:** 2
**Contribution:** 2
**Rating:** 4
**Confidence:** 3

**Summary:**

This paper proposed a framework for LLM-based grounded theory development.

**Strengths:**

Address grounded theory computationally is a novel direction.

**Weaknesses:**

1. Lack of justification on the 5-D evaluation of code quality. Why are these chosen? Some recent work has leverage computational version of principle of trustworthiness to evaluate qualitative results, what is the merit of the proposed evaluation method compared to existing ones?
2. Lack of discussion and comparison with existing computational qualitative analysis methods with LLMs, in related work and experiment.
3. Lack of details and experimental analysis on embedding methods, clustering methods and graph construction, as different choices within each method will affect results. This also hinders reproducibility.

**Questions:**

See weakness.

---

> ### Author Response · Authors · 2025-11-25
> **Response to Reviewer a3QD (Part 1 / 2)**
>
> ### 1.  Justification of the 5-D metric and relation to “trustworthiness”
>
> We appreciate the reviewer’s question about why we chose this particular 5-D evaluation and how it relates to existing notions of “trustworthiness” in LLM-based qualitative analysis.
>
> Our goal in this paper is to **evaluate the quality of an induced codebook as an analytic artifact**, not to evaluate the intrinsic trustworthiness of an LLM model. In the existing literature, “trustworthiness” is used in several distinct ways, none of which directly match this evaluation target:
>
> 1. **Model-centric trustworthiness (e.g., TrustLLM).**
>    Much recent work (e.g., *TrustLLM*) defines trustworthiness as a property of the *model* itself, with dimensions such as truthfulness, safety, fairness, robustness, privacy, transparency, accountability, and machine ethics. These are crucial when evaluating LLMs as deployed systems, but they are **not properties of a static codebook**. Our metric evaluates the *output schema* (the set of codes and their relations) with respect to the corpus and research question. A codebook cannot, by itself, “leak privacy” or “emit toxic content” in the way a generative model can. Applying these model-centric trust metrics to a codebook would be a category error.
>
> 2. **Uncertainty-based trustworthiness (perplexity / confidence).**
>    Another technical notion equates trust with uncertainty estimates such as perplexity. This is inappropriate for our purpose: a codebook is a **hierarchical system** of many codes, not a single generated sentence. Aggregating perplexities over hundreds of codes would not tell us whether the resulting schema is reusable, non-redundant, or well-aligned with the corpus; it does not capture the holistic quality of the codebook.
>
> 3. **Human–AI alignment as trustworthiness.**
>    In qualitative methods, trustworthiness is often framed via credibility, dependability, and confirmability (e.g., Lincoln & Guba; see “Practical guidance to qualitative research. Part 4: Trustworthiness and publishing”, 2021). Recent computational work on LLMs in qualitative analysis (e.g., Bennis et al., 2025; Oksanen et al., 2025; Zhao & Liu, 2025) shows that LLM-generated codes can align well with human coding and be consistent across runs. Our own human-alignment studies (MAS, YC, Vibe Coding) likewise show high agreement with expert schemas. However, **human alignment alone is not sufficient as a quality metric for codebooks**: human teams can also produce codebooks that are overly specific, redundant, or weakly generalizable. Our aim is not just to mimic human annotators, but to assess the *quality of any codebook* (human- or AI-generated) against the data.
>
> ### 2. Justification on why we use a 5-D codebook metric
>
> Our 5-D metric is explicitly derived from the “Five Highest Desiderata of Codebooks” section (which we have clarified in the revision). We are not measuring “trustworthiness of the LLM” in the abstract, but **codebook quality with respect to the corpus and research question**. The five dimensions are:
>
> - **Reusability** – do codes actually get reused across datapoints, enabling generalization and cross-case analysis?
> - **Descriptive Fitness** – do the assigned codes accurately describe each datapoint?
> - **Descriptive Coverage** – do the codes capture all essential aspects of a datapoint relative to the research question?
> - **Parsimoniousness** – does the codebook avoid redundant, semantically overlapping codes?
> - **Consistency / Stability** – does the code distribution remain stable between train and test splits (i.e., does the schema generalize to unseen data)?
>
> These dimensions directly operationalize what qualitative researchers need from a codebook: a schema that is **generalizable, accurate, comprehensive, compact, and stable**. They are also naturally related to traditional trustworthiness criteria:
>
> - Fitness and coverage relate to **credibility**.
> - Reusability and stability relate to **transferability** and **dependability**.
> - Parsimony supports **analytic clarity** and **confirmability**.
>
> Thus, our metric does not ignore trustworthiness; it **recasts it at the appropriate level of analysis**. Rather than evaluating the LLM as a conversational agent, we evaluate the *codebook* it produces. The 5-D metric is designed to be:
>
> 1. **Model-agnostic** – applicable to codebooks produced by humans or by LLMs.
> 2. **Structure-aware** – capturing semantic and structural properties of the schema, not just properties of individual responses.
> 3. **Complementary** to traditional qualitative judgment – providing an objective, quantitative lens that can sit alongside human evaluation and the broader trustworthiness discourse.
>
> We have updated the “Five desiderata” section to make this connection explicit and to emphasize that our 5-D metric is intended as a principled, codebook-level operationalization of quality, rather than a replacement for broader model-centric notions of LLM trustworthiness.

---

> ### Author Response · Authors · 2025-11-25
> **Response to Reviewer a3QD (Part 2 / 2)**
>
> ### 3. Comparison with existing computational qualitative methods
>
> **Comment:**
> No comparison with existing computational qualitative analysis methods using LLMs, in related work or experiments.
>
> **Response:**
> We have already incorporated discussion of the most relevant LLM-based qualitative systems (e.g., CoAICoder, CollabCoder, MindCoder, AcademiaOS, LLOOM) in the **Related Work** section and highlighted their limitations relative to LOGOS. These systems typically:
>
> - support **assisted coding** (LLM-as-collaborator or suggestion engine), or
> - produce **flat or lightly structured themes**, often requiring substantial expert curation and validation.
>
> In contrast, LOGOS:
>
> - induces a **hierarchical, relation-aware codebook** (with subsumption, equivalence, and orthogonality),
> - includes an **iterative refinement loop** to optimize reusability and abstraction level, and
> - is evaluated under a **train–test protocol** with the 5-D metric described above, plus **human-alignment studies** on MAS, YC, and Vibe Coding. Please see **our additional studies  in the Appendix section.**
>
> We will make this contrast more explicit in the related work section and note that our evaluation framework is designed to be *generic*: it could also be applied to codebooks produced by these prior systems, providing a common basis for future comparison.
>
> (For completeness, we maintain an internal literature sheet tracking existing computational qualitative analysis methods; we would be grateful for any additional pointers the reviewer may have.)
>
>
> ### 4. Details and experimental analysis on embeddings, clustering, and graph construction
>
> **Comment:**
> Lack of details and experimental analysis on embedding methods, clustering methods and graph construction, as different choices within each method will affect results. This also hinders reproducibility.
>
> **Response:**
> We agree that different component choices can influence behavior, and we will clarify our design motivations and implementation details more explicitly in the paper. Our philosophy, following GraphRAG and LightRAG, is that the primary contribution is the **end-to-end framework** for autonomous grounded-theory analysis, with components chosen to be:
>
> - strong but **standard** (for ease of reproduction), and
> - easily **swappable** (so practitioners can plug in alternative embeddings or clusterers if desired).
>
> Specifically:
>
> - **Embeddings.** We use `qwen3-embed-0.6B` as a modern, high-quality embedding model. Prior work on document retrieval and topic clustering suggests that substituting contemporary embedding models (e.g., BERT, SBERT, E5) typically yields **only marginal improvements** (often within 1–3% in clustering coherence or retrieval F1; see e.g., Reimers & Gurevych, 2019; Wang et al., 2023). These small differences are likely to be diluted through the rest of the LOGOS pipeline, which includes open coding, relation classification, and iterative refinement.
>
> - **Clustering.** We deliberately adopt **K-means** with Silhouette/variance-based selection of K. Density-based methods such as HDBSCAN or DBSCAN assume well-separated dense regions, but our code embeddings live in a high-dimensional semantic space that is relatively sparse and diffuse. In such regimes, HDBSCAN/DBSCAN tend to assign many points to the “noise” label (-1), which is problematic when we later rely on cluster membership for relation classification and graph construction. K-means, by contrast, provides stable, deterministic assignments for all codes and integrates cleanly with the rest of LOGOS.
>
> - **Graph construction.** We fully specify our graph construction rules in the Method section: we classify code pairs using a distilled relation classifier, filter by cosine similarity thresholds, and then apply explicit **deductive closure** rules (transitivity and equivalence) to build the conceptual graph. Conflicts are resolved via a “deduction-first” principle. We will add a few more concrete implementation details (e.g., similarity thresholds, top-% selection) to ensure reproducibility.
>
> Finally, we plan to release LOGOS as a **Python library** with clear documentation. Because the framework is fully modular, its core components (embedding model, clustering algorithm, similarity thresholds, relation classifier) can be reproduced, replaced, or extended by future work. Our intent is that LOGOS defines the **algorithmic scaffold**, while allowing the community to experiment with alternative components within this scaffold.

---

> > ### Comment · Reviewer_a3QD · 2025-11-26
> >
> > The top issue I have is still the lack of comparison with existing work; there is a lot of recent literature that can be found by just searching a few key words, such as ''thematic analysis LLM'', "qualitative LLM". Reviewer RhmW has listed a few that you have not compared with either in related work or in the experiment. Among these, one of the works has proposed evaluation metrics based on the principle of trustworthiness, which should also be compared with your evaluation metrics, through experiment, not by arguing that your proposed metrics are more suitable, even though your proposed metrics lack support from qualitative research literature. Regarding those mentioned in related work, no experiments were conducted to compare them either.
> >
> > The second most concerning point is the lack of technical detail, which hinders reproducibility. Although the promise of releasing an open library is great, the paper and the response still lack enough technical details to reproduce the results, such as hyperparameter settings. Choices of encoder and clustering methods are not justified or compared with similar methods.

---

> > > ### Author Response · Authors · 2025-11-29
> > > **Response to Reviewer a3QD - Round 2**
> > >
> > > ### 1. Literatures and Comparison
> > > **Comment**:
> > >
> > > The top issue I have is still the lack of comparison with existing work…no experiments were conducted to compare them either.
> > >
> > > **Response**:
> > >
> > > We have **added 20+ related literature in the related work section** discussing the difference and shortcoming of these previous works compared to LOGOS. Most notably, we perform experiments on **Thematic-LM (WWW 2025) and HICode (EMNLP 2025)** across all of our datasets (Ali, Podcast, Paper Abstract, MAS, Math Failure). Across all datasets, **LOGOS performs on par or better with these competent methods**, and we have updated the new results in Table 1 and Figure 2 in the manuscript for comparison.
> > >
> > > In regard to the metric, we also compare between LOGOS proposed 5 Levels Metric vs Thematic-LM metric. Both LOGOS and Thematic-LM aim to evaluate three core properties of an induced coding/theming pipeline: (1) how well the learned structure fits the data, (2) how stable it is, and (3) how well it generalizes. The key difference is how they operationalize these goals.
> > >
> > > **(1) Fitness + coverage ↔ credibility / confirmability**
> > >
> > > Thematic-LM relies mainly on ROUGE overlap and a single LLM “consistency” score. Its credibility/confirmability metric aggregates a binary judgment of whether each theme matches its associated quote, but this signal is coarse and is not validated against human raters. It also defines stability as run-to-run reproducibility under stochastic sampling, which is sensitive to phrasing differences and decoder randomness.
> > > LOGOS instead uses semantic, prediction-style metrics, and evaluates a fixed learned schema on held-out data. It also validates these metrics directly against expert qualitative judgments. LOGOS decomposes credibility/confirmability into two interpretable components:
> > > - **Fitness**: how well individual codes/themes align with specific data instances.
> > > - **Coverage**: how much of the dataset is meaningfully accounted for by the schema.
> > > These capture per-item fit and corpus-level breadth separately, rather than merging both into a single score. Importantly, both correlate strongly with human judgments on held-out examples.
> > >
> > > **(2) Consistency + reusability ↔ transferability (and about ROUGE)**
> > >
> > > For stability and generalization, Thematic-LM uses ROUGE-1/2 between theme descriptions across runs or splits, which **largely measures lexical overlap, not conceptual similarity**. Semantically identical themes with different wording can score low (e.g.  “Eco-anxiety about climate change” vs. “Climate-related concerns and worries”), while boilerplate rephrasing can score high.
> > > LOGOS avoids this problem in two ways:
> > > - **Consistency (1 − JSD)**: compares the distribution of code usage across train/test splits, independent of wording.
> > > - **Reusability**: tests whether a codebook learned on the train split can be reapplied to held-out data without inventing new codes, measuring how many codes remain useful and how well they fit and cover unseen examples.
> > >
> > > Together, these give a stronger notion of transferability: not “Do two runs produce similar text strings?” but “Does a fixed schema remain meaningful and stable on new data?”
> > >
> > > **(3) Why we do not report dependability (in Thematic-LM’s sense)**
> > >
> > > For fairness of comparison, we have fixed a single LLM (Qwen3-32B) with **temperature = $0.0$** for methods through our experiments. With greedy decoding, **all methods are deterministic**. Thus repeated runs are yield **identical** results, making dependability **trivially maximal and uninformative**. LOGOS instead emphasizes distributional stability (consistency) and schema reuse (reusability), which capture robustness to changes in data rather than randomness in generation.
> > >
> > >
> > >
> > > ### 2. Implementation Technical details
> > > **Comment**:
> > >
> > > The second most concerning point is the lack of technical detail…not justified or compared with similar methods.
> > >
> > > **Response**:
> > >
> > > The technical detail for LOGOS has been added in the Appendix section and will open source LOGOS in the future.

---

### Official Review · Reviewer_8qug · 2025-10-31

**Soundness:** 2
**Presentation:** 3
**Contribution:** 3
**Rating:** 8
**Confidence:** 4

**Summary:**

Authors introduce a new method to automate grounded theory qualitative coding. Their method outperforms other baselines according to a variety of metrics. In addition, it is validated on a dataset of expert-identified problems in multi-agent reasoning.

**Strengths:**

- Well-motivated and theoretically grounded study. The method makes sense, is clearly described, and every part has grounding in how coding is done in qualitative studies. The paper also explains the background very well, clearly outlining steps for the grounded theory annotation. Finally, the paper explains how these methods are actually pervasive (unknowingly) in our field. If it is easily reproducible, I am sure many AI researchers could greatly benefit from using this method for error analysis (instead of the current approaches that are much more ad-hoc).

- Thorough evaluation: sound statistical metrics to measure the proposed method's performance (though I have some questions about LLM as a judge for some components) and its advantage over baselines. Representative and strong baselines were selected for comparison. However, there are issues with not reporting statistical significance.

- Case studies provide examples of how the method can be successfully used in such pertinent applications as math reasoning failures identification, multi-agent system debugging.

**Weaknesses:**

- Some evaluation is done with LLM judges with no validation of their accuracy (specifically, descriptive fitness and coverage).

- Summed metrics are reported, but the metrics are on different scale (eg 1-10 vs 0-1). That would give some metrics unequal weight, so they should be normalized before summing.

- Table 1 does not report statistical significance. At the very least standard deviation should be reported. Ideally some tests should be performed to get p values or report confidence intervals.

- Case study in experiment 3 was quite confusing. Why was the result from LOGOS acceptable but not from the other two systems? What was the ideal result supposed to look like?

- Lack of a limitation/ethics section. There could be ethical issues with LLMs misinterpreting / interpreting in a biased way the qualitative data in a way experts would not, especially in qualitative data from human subjects. The authors should identify and discuss such limitations.

**Questions:**

> In some sense, even though there are admittedly many abundant back-and-forth iterations, conceptual and theoretical framework are usually developed first, then computer scientists embody and animate the static-metaphysical conceptualization by programming workflow.

This sentence is very dense and unclear. What did the author actually mean to say?

line 76: strives -> strive

>because flexible aggregation and multi-view clustering (e.g. via logic operations) relies on high code-sharing rate across datapoints

this justification is very unclear

- Section 3.3 is unclear. What is the code prediction step "similar to the codebook update in iterative refinement procedure"?

---

> ### Author Response · Authors · 2025-11-25
> **Response to Reviewer 8qug (Part 1 / 3)**
>
> ### 1. LLM judge accuracy for fitness and coverage
>
> **Comment:**
> Some evaluation is done with LLM judges with no validation of their accuracy (specifically, descriptive fitness and coverage).
>
> **Response:**
> We agree that validating LLM-judged metrics is important. During development we conducted a **pilot human rating study** on the Math Failure dataset:
> We sampled $(n = 100)$ datapoints (text–code pairs) and obtained human ratings for both descriptive fitness and coverage using the same 1–10 rubric as the LLM. We then compared these with the LLM’s scores and found high correlations: `Pearson` ($r = 0.73$) for fitness and ($r = 0.70$) for `coverage`. This indicates that the LLM judgments for these metrics are strongly aligned with human judgments, supporting their use as proxies in our evaluation.
> We have now included this pilot study and its results in the Appendix of the revised paper to demonstrate the reliability of our automated evaluation.
>
> ### 2. Different metric scales and normalization before summing
>
> **Comment:**
> Summed metrics are reported...They should be normalized before summing.
>
> **Response:**
> We agree, and we have already normalized the metrics in the revised text:
>
> * **Descriptive fitness** and **descriptive coverage** are originally scored by the LLM on a **1–10** scale.
> * We now **rescale both to ([0,1])** before computing the composite score, to make them consistent with the other dimensions (reusability, parsimony, stability).
>
> Concretely, in Section 3.3 we now state:
>
> ```latex
> Descriptive Fitness. A score assigned by a large language model (LLM) in the range of 1--10, that \textit{quantifies the extent to which selected codes properly describe a datapoint (even if partially)}. For example, describing a restaurant review that complains about delivery speed as “parking difficulty” would be considered improper. To keep the scale consistent with other metric dimensions, we rescale the fitness score to [0, 1].
> ```
>
> and
>
> ```latex
> Descriptive Coverage. Similarly LLM-scored in the range of 1--10, coverage evaluates to what extent all the essential aspects of a datapoint relevant to the research question are captured by the assigned codes. ... We also rescale the coverage score to [0, 1] to keep magnitude consistency.
> ```
>
> The composite metric in Table 1 is then computed from **five values all in ([0,1])**.
>
>
> ### 3. Lack of statistical significance / variance reporting in Table 1
>
> **Comment:**
> Table 1 does not report statistical significance. Standard deviation, p-values, or confidence intervals should be reported.
>
> **Response:**
> We appreciate the suggestion. In our setting, however, the usual variance sources are largely suppressed by design, and additional random restarts would be extremely expensive for limited benefit.
>
> 1. **Near-deterministic pipeline.**
>    Across all experiments we use:
>
>    * very low decoding temperatures (**0 or 0.1**),
>    * fixed prompts and hyperparameters for all methods.
>
>    Given a fixed corpus, each pipeline (including baselines) produces essentially the **same codebook and metrics** on every run. The main remaining randomness is from, e.g., `k`-means initialization, which has negligible impact at our corpus sizes.
>
> 2. **Corpus-level, not small-sample metrics.**
>    The metrics in Table 1—reusability, parsimony, and stability—are computed over the **entire test corpus**, not tiny subsamples, so they are already close to population-level quantities rather than noisy estimates.
>
>    For the LLM-judged **fitness** and **coverage** metrics, we use a reasonably large random sample of datapoints from the test set and report single-point estimates; we will clarify this sampling protocol in the revision.
>
> 3. **Robustness via multiple datasets and research questions.**
>    Instead of relying on variance over random seeds on a single dataset, we assess robustness by evaluating on **five diverse corpora**, each with multiple research questions. LOGOS consistently ranks best under the composite metric and shows similar advantages in the per-dimension breakdown (Figure 2). We view this **cross-dataset consistency** as a stronger robustness signal than small variations across multiple seeds on one dataset.
>
> 4. **Computational and environmental cost.**
>    Each full LOGOS run involves:
>
>    * multiple LLM passes over the entire corpus,
>    * relation classification and graph construction, and
>    * several refinement iterations.
>
>    Re-running every method many times per dataset to estimate variance would be **prohibitively expensive**, and would incur substantial compute and environmental cost, for very limited additional insight given the near-deterministic behavior above.
>
> In the revised paper, we will explicitly:
>
> * describe the **low-temperature, fixed-prompt** setup,
> * clarify the **sampling protocol** for LLM-based metrics,
> * state that we therefore report **single-run corpus-level scores** in Table 1.

---

> ### Author Response · Authors · 2025-11-25
> **Response to Reviewer 8qug (Part 2 / 3)**
>
> ### 4. Experiment 3 case study (LOGOS vs GraphRAG / LightRAG)
>
> **Comment:**
> Case study in Experiment 3 is confusing. It is unclear why the result from LOGOS was acceptable but not from the other two systems, or what the ideal result looks like.
>
> **Response:**
>  The intent of Experiment 3 is to evaluate **global sense-making / schema-induction behavior**, not ordinary QA on a single item.
>
> * The corpus is the **Math Failure dataset**, and the global research question is:
>
>   > *“What are LLMs’ recurring failure patterns in mathematical reasoning?”*
>
> * An acceptable answer must:
>
>   1. Identify **recurring failure modes** (e.g., misread constraints, misapplied arithmetic operations, unit inconsistencies, wrong aggregation logic, missing validation), and
>   2. Synthesize them into a **corpus-level taxonomy**, rather than describing only a single example.
>
> Under this criterion:
>
> * **GraphRAG**’s answer explicitly claims that the dataset “contains no explicit math problems, proposed solutions, or incorrect results… so failure analysis isn’t possible.” This is factually incorrect—the dataset *does* consist of math problems with incorrect LLM solutions—and it **produces no failure patterns at all**. As a result, it simply fails to answer the schema-induction question.
>
> * **LightRAG** focuses on a single example (“Lou ate half the cookies”) and explains the contradiction in that specific story, but **does not generalize** to corpus-level failure patterns. It behaves like standard RAG explaining an instance, not like a system inducing a global error taxonomy.
>
> This behavior reflects their design: both build entity/relation graphs and focus on **surface-level semantic links**. They do not represent the **internal reasoning steps** (e.g., how constraints are interpreted, where arithmetic is misapplied, whether validation is performed), so the graph lacks the key information needed to induce error categories.
>
> **LOGOS**, by contrast:
>
> * performs **LLM-based open coding** at the datapoint level, explicitly interpreting each incorrect solution (e.g., “misread constraint,” “added instead of multiplied,” “no validation against initial condition”),
> * clusters and relates these codes via its relation-aware graph,
> * and then produces a **taxonomy of recurring math failure patterns**.
>
> The answer we show for LOGOS in Figure 4 enumerates errors such as:
>
> * misread/misused constraints,
> * misapplied arithmetic operations,
> * confusion between multiplicative vs. additive reasoning,
> * inconsistent units and totals,
> * missing validation loops,
> * cumulative logical breakdown leading to contradictions with the problem context.
>
> This is exactly the kind of **global error taxonomy** we aim to evaluate.
>
> ### 5. Lack of limitations / ethics section
>
> **Comment:**
> Lack of a limitation/ethics section. Ethical issues may arise from LLMs misinterpreting or biasing qualitative data.
>
> **Response:**
> We agree and have added a dedicated **Limitations and Ethical Considerations** section. We also want to note that we primarily apply LOGOS to **epistemic/technical corpora** (e.g., math failures, multi-agent traces, research talks, YC pitches), where the main risk is *epistemic mischaracterization* of patterns, not direct social/identity labeling. For more sensitive social-science applications, we explicitly recommend that LOGOS be used strictly **with human-in-the-loop review** of the full codebook.
>
> ### 6. Dense and unclear sentence about conceptual frameworks and workflows
>
> **Comment:**
> The sentence starting “In some sense, even though there are admittedly many abundant back-and-forth iterations…” is dense and unclear.
>
> **Response:**
> We agree and have replaced it with a clearer formulation:
>
> ```latex
> Typically, computer scientists first derive the conceptual categories and theoretical structure through extensive manual iteration, and only later convert this understanding into a coded automated annotation workflow. LOGOS aims to automate this otherwise conceptual and labor-intensive step.

---

> ### Author Response · Authors · 2025-11-25
> **Response to Reviewer 8qug (Part 3 / 3)**
>
> ### 7. Typo Error
> Have already fixed in our paper.
>
> ### 8. Unclear justification for reusability / multi-view clustering
>
> **Comment:**
> “because flexible aggregation and multi-view clustering (e.g. via logic operations) relies on high code-sharing rate across datapoints — this justification is very unclear.”
>
> **Response:**
> We have simplified and clarified this desideratum in Section 2. The updated text now reads:
>
> ```latex
> (1) maximally reusable and generalizable: the same code should apply to many different datapoints, so that researchers can flexibly aggregate and slice the corpus by shared codes (e.g., using Boolean combinations such as “all datapoints with codes A AND B but not C”);
> ```
>
> This keeps the original intent (support for multi-view clustering and logic-based subsets) while making the intuition much more concrete.
>
> ### 9. Section 3.3 “code prediction step” unclear
>
> **Comment:**
> Section 3.3 is unclear: What is the code prediction step “similar to the codebook update in iterative refinement procedure”?
>
> **Response:**
> We agree that this was underspecified and have clarified that the test-set coding is **deductive**, using the train-derived codebook and the same retrieval mechanism as in refinement. The revised paragraph in Section 3.3 reads:
>
> ```latex
> Following the standard train–test-split protocol in machine learning, with codebook fitting done on the train set, we perform code prediction on the test set via a retrieval-based selection algorithm similar to the codebook update in the iterative refinement procedure: for each test datapoint we retrieve a candidate pool of existing codes using the same semantic+graph retrieval as in Section~3.2, and an LLM then selects (but is not allowed to invent) up to 20 codes from this pool. In other words, the test set is coded deductively using only the train-derived codebook.
> ```
>
> We also clarify the **Reusability** metric to make use of “used codes” precise:
>
> ```markdown
> Reusability. Defined as the ratio of used codes to all codes:
> $$
> \text{Reusability} = \frac{\#\text{used codes}}{\#\text{all codes}}
> $$
> Here, *all codes* refers to the size of the final, cleaned codebook learned on the train split (after Steps 4–6 and iterative refinement). A *used* code is any code from this train-derived codebook that is assigned at least once to a test datapoint during the deductive code prediction step described above; codes that remain in the codebook but are never selected on the test set are counted as unused.
> ```
>
> Overall, we thank Reviewer 1 for these detailed comments. They helped us clarify our methodology (especially the evaluation protocol), tighten the writing, and make our assumptions and limitations more explicit in the revised manuscript.

---

### Author Response · Authors · 2025-11-29
**Note to AC (Part 1 / 2)**

We sincerely appreciate the reviewers’ thoughtful engagement.

LOGOS introduces an end-to-end LLM-based framework that automates key steps of inductive coding—open coding, clustering, and abstraction—while providing quantitative metrics to assess and refine the resulting codebooks. Several reviewers highlighted core strengths of the work: a **systematic, theoretically grounded framework** for automating grounded-theory-inspired coding, along with **novel metrics, strong baselines**, and **comprehensive evaluations across diverse datasets**.

Reviewer 1 described the study as *“well-motivated and theoretically grounded,”* noting that *“every part of the method is grounded in how coding is done in qualitative studies,”* and that the background and procedures are *“clearly explained.”* Reviewer 3 emphasized that our proposed metrics *“cover a wide range of important concerns,”* addressing an area with few established standards, while Reviewer 4 noted that the paper *“tackles a major bottleneck”* by providing an *“end-to-end LLM workflow”* for inductive coding, clustering, and iteration. We are grateful for this recognition.

# A Deeper, Under-Highlighted Contribution: A New Possibility for AI Error Analysis

Beyond the technical pipeline and metrics, there is another key contribution of this work that may be easy to overlook if one focuses only on the mechanics of LOGOS. A central motivation behind the framework is to show that qualitative research methodology — specifically grounded theory — can be brought into the context of large-scale AI systems through a largely automated workflow. Our work provides an initial demonstration that such qualitative-inspired analysis can be systematically applied to AI-generated outputs at scale.

 > **To the best of our knowledge, this is the first work in computer science to adapt grounded-theory methodology for the automated error analysis of large-scale, AI-produced answers and complex model behaviors.**

This serves as an early feasibility result, indicating that qualitative and inductive coding practices can be meaningfully operationalized within computational pipelines.

We do not claim to fully automate grounded theory or solve error analysis in its entirety. Rather, we aim to demonstrate that the core ideas—open coding, axial organization, iterative refinement—can be meaningfully operationalized in an automated framework, and that this approach can scale to realistic AI system outputs.

This broader contribution extends beyond LOGOS as a specific instantiation. The method is intended as a general framework: a structured way to turn complex system failures into quantitative, analyzable codebooks and graphs.

> **As Reviewer 1 noted: “If it is easily reproducible, I am sure many AI researchers could greatly benefit from using this method for error analysis (instead of the current approaches that are much more ad-hoc)”**

**We will prepare LOGOS as a publicly accessible python package that can be easily pip-installed**.


The empirical results support this direction across multiple settings. The **Multi-Agent System (MAS) failure** case study offers one example of large-scale, complex error analysis (i.e. in the optimal setting, reaching **88.2%  human alignment with human manual checking**, with $80.4%$ on average). LOGOS also generates insightful taxonomies for **LLM math reasoning failures** and even for the **scientific paper abstracts analysis**, suggesting that the same workflow applies across meaningfully different domains. Because LOGOS operates on coded representations, it can also extend to other modalities—**such as images or audio**—simply by modifying the open-coding step.

## Why This Matters
As stated in our introduction, recent MAS and large-model debugging studies indicate that current error analysis practices often rely on **extensive manual annotation**, domain-specific heuristics, and non-systematic exploratory “poking.” This makes large-scale behaviorial pattern and failure analysis time-consuming and difficult to replicate.

Our results suggest that this process can instead be approached with a **structured, automated, and extensible framework like LOGOS**. This shift can: (1) support more systematic exploration of failure modes; (2) enable more quantitative comparison across systems; and (3) foster interdisciplinary collaboration, by making grounded-theory-style analysis accessible to AI researchers and familiar to qualitative-methods researchers.

In this sense, LOGOS may bring  **a paradigm shift to how CS/AI researchers think about and approach model behavioral pattern and error analysis**, by showing that practices once viewed as qualitative and artisanal can become more scalable, reproducible, and broadly applicable.

> **We humbly hope the AC could consider our contribution in revealing the broader possibility that automated, structured error analysis can become a standard and scalable tool in AI research.** Thanks!

---

> ### Author Response · Authors · 2025-11-29
> **Note to AC (Part 2 / 2)**
>
> In response to the reviewers' insightful comments and concerns, we made substantial updates to both the manuscript and the supplementary experiments. Below is a consolidated summary of improvements, grouped by major concern categories and tied directly to reviewer requests.
>
> ## 1. Expanded Related Work & Clearer Positioning
> `Addresses Reviewers a3QD & RhmW`
> We significantly strengthened the Related Work section by adding **20+ papers on qualitative analysis**, including:
> Computational automation including: HICode, Thematic-LM, MindCoder, CollabCoder, CoAICoder, Auto-TA, ThemeViz, etc.
> Qualitative Research Principle Papers:  Practice and process in qualitative research (thematic analysis, grounded theory and their comparison), Evaluation of qualitative research (trustworthiness–based evaluations, and description coverage and fitness)
> Domain Specific Application Paper: Theoretical schema induction for Literature surveys and software engineering
>
>
> We also added a **conceptual distinction** between: **thematic analysis pipelines** (focused on theme summaries), and **grounded theory pipelines** (focused on hierarchical relational codebooks and theory-building).
>
>
> ## 2. Added Two New Strong Baselines: HICode & Thematic-LM
> `Addresses Reviewer a3QD & RhmW`
> We incorporated **HICode (EMNLP 2025)** and **Thematic-LM (WWW 2025)**, the two most recent and competent LLM-based inductive thematic analysis workflows.
> We adapted or reimplemented both systems faithfully and added them to all datasets.
> **Result**: LOGOS performs **on par or better** than both across all five corpora, confirming that LOGOS is strongly competitive even against the strongest TA systems.
>
> ## 3. Validation of LLM-as-a-Judge
> `Addresses Reviewers 8qug & tzNx` We added a **human study** to validate LLM judgments of fitness and coverage:
> - **Pearson r = 0.73** for fitness.
> - **Pearson r = 0.70** for coverage.
> This demonstrates that LLM scores correlate strongly with human scores.
>  The entire procedure (sampling protocol, prompts, human rubric) is added to **Appendix A.3.3**.
>
> ## 4. Two Additional Human-Alignment Studies (YC + Vibe Coding)
> `Addresses Reviewer tzNx` Beyond the MAS dataset, we added:
> - YC 60-second founder pitch dataset with human-coded ground truth, where LOGOS achieves **80% alignment**. Corpora + codebook uploaded to supplimentary material.
> - Vibe Coding dataset (Sarkar & Drosos 2025) based on 33 YouTube transcripts, where LOGOS achieves **100% alignment**, verified manually
>
> **Reviewer tzNx (the most critical 2-rating reviewer)** explicitly expressed satisfaction with these added studies.
>
> ## 5. Ethics & Limitations Section Added
> `Addresses Reviewers 8qug & tzNx`
> The revised paper now includes a full Ethics/Limitation section in Appendix addressing:
>
> - LLM hallucination and epistemic misinterpretations
> - Biases in coding or schema construction
> - Risks of fully autonomous GT pipelines
> - The need for human-in-the-loop review for sensitive domains
> - Limitations of semantic-only code relations (temporal, causal, procedural relations remain future work)
>
> Reviewer tzNx specifically requested cautionary statements about **factual errors unrelated to bias**, which we explicitly added.
>
> ## 6. Additional Technical Detail for Reproducibility
> `Addresses Reviewers a3QD & tzNx`
> We added:
> - All **hyperparameters** (LR, batch sizes, temperatures, thresholds)
> - Complete **embedding, clustering, and graph construction settings**
> - Explanation of classifier training and balancing strategies
> - Detailed description of: candidate generation, retrieval ranking, refinement iteration, graph conflict resolution
>
> These are included in Appendix A.7. We also improved the details in the Method section.
>
> ## 7. Improved Writing, Symbolic Notation & Clarity
>
> We performed a full editing pass:
> - corrected all typos (e.g., “developemt”, “extreemly”).
> - clarified symbolic definitions of **R, used codes, Ssemantic**, etc.
> - clarified the inductive vs deductive cycle in LOGOS.
> - clarified the purpose and fairness of codebook aggregation.
> - restructured ambiguous sentences.
>
> ## Final Note to AC
> Across the rebuttal period, we have:
>
> - **Addressed every reviewer concern** with concrete improvements and empirical results.
> - **Added new experiments, baselines, and human studies**—significantly strengthening both empirical and methodological justification.
> - **Clarified the conceptual positioning** vs. thematic analysis, which was a central concern from multiple reviewers.
> - Brought the paper to a level that the most critical 2-rating reviewer (tzNx) described as “substantially improved.”
>
> We believe the revised submission now presents a **technically solid and well-validated contribution**, offering a novel fully automated grounded theory development framework with: strong baselines, strong human alignment, validated metrics, and clear conceptual novelty.

---

### Meta-Review · Area_Chair_688V · 2025-12-09

**Summary:**

The authors made a substantial effort to address reviews. In the current situation, where the real people that looked at it in depth are away, I am left with an unfair decision. Practically, among the weaknesses were overclaiming, unfactual comments, unsubstantiated claims and a lot of missing writing. Due to all that, despite their addition during a rebuttal, it is likely that a lot of other things are not addressed, that some are not addressed well enough being rebuttal changes etc.
Regarding the missed point across reviewers, I suggest reframing the paper to ensure that the paper's points are clearly conveyed (and yes, when things are more complicated spoonfeeding is necessary, but the same would be in people reading the paper, so if reviewers not get it, readers won't as well.)

**Reviewer Concerns:**

Among the concerns were claims with no experiments to support them (e.g. usability to replace human annotators), overclaiming or substandard methodology (off-handed LLM as a judge).

**Reviewer Scores:**

That is not a fair, relevant or meaningful question. I protest the way this was all handled.
A Reviewers are not here, and ToM is weak, at least mine and the one literature study. I will not try to predict people.
B Scores are anyway a weak signal of interest, a paper should not accepted or rejected just based on it, that's an ACs job to look at the specific weaknesses and translate it into a recommendation.
C There are about 100 pages of discussions for me to read overall, in addition to the discussions I monitored and were just replaced, this is beyond my personal ability to do fairly. I did my best effort.

---

### Decision · Program_Chairs · 2026-01-26

Reject